,

# BrAHMs V1.0: A fast, physically-based subglacial hydrology model for continental-scale application.

Mark Kavanagh[1] and Lev Tarasov[2]

[1]Faculty of Engineering and Applied Science, Memorial University of Newfoundland, St. John's, NL, Canada
[2]Dept. of Physics and Physical Oceanography, Memorial University of Newfoundland, St. John's, NL, Canada

*Correspondence to:* Lev Tarasov
(lev@mun.ca)

**Abstract.** We present BrAHMs (BAsal Hydrology Model): a physically-based basal hydrology model which represents water flow using Darcian flow in the distributed drainage regime and a fast down-gradient solver in the channelized regime. Switching from distributed to channelized drainage occurs when appropriate flow conditions are met. The model is designed for long-term integrations of continental ice sheets. The Darcian flow is simulated with a robust combination of the Heun and leapfrog-trapezoidal predictor-corrector schemes. These numerical schemes are applied to a set of flux-conserving equations cast over a staggered grid with water thickness at the centres and fluxes defined at the interface. Basal conditions (e.g. till thickness, hydraulic conductivity) are parameterized so the model is adaptable to a variety of ice sheets. Given the intended scales, basal water pressure is limited to ice overburden pressure, and dynamic time-stepping is used to ensure that the CFL condition is met for numerical stability.

The model is validated with a synthetic ice sheet geometry and different bed topographies to test basic water flow properties and mass conservation. Synthetic ice sheet tests show that the model behaves as expected with water flowing down-gradient, forming lakes in a potential well or reaching a terminus and exiting the ice sheet. Channel formation occurs periodically over different sections of the ice sheet and, when extensive, displays the arborescent configuration expected of Röthlisberger Channels. The model is also shown to be stable under high frequency oscillatory meltwater inputs.

## 1 Introduction

Subglacial basal hydrology is a potentially critical control on basal drag and therefore ice streaming. It is also a clear control for subglacial sediment production/transport/deposition processes (Benn and Evans, 2010; Melanson, 2012). Subglacial water flows can also leave clear geological imprints. For instance, eskers are a geological footprint of past channelized subglacial drainage (Benn and Evans, 2010) that can in turn be used to better constrain past ice sheet evolution.

Many models relating to basal hydrology are either meant for short time scales (e.g., on the order of weeks to centuries), or are missing a key component of basal water flow (channelized flow). We present a computationally-fast physics-based subglacial hydrology model for continental-scale ice sheet systems modelling over glacial cycles that is meant to capture the relevant features of basal water flow for the above three contexts (including both distributed and channelized flow components).

This large spatio-temporal scale context places a high requirement on computational speed and justifies certain simplifications compared to glacier-scale models (Bartholomaus et al., 2011; Werder et al., 2013; de Fleurian et al., 2016). Glacial cycle models do not resolve daily or even weekly mean changes in basal drag and spatial scales are relatively coarse (10-50 km). As such, the detailed physics of cavity evolution and tunnel formation cannot be resolved (given their dependence on basal sliding velocities) nor, we posit, need they be resolved. The latter is justified on the large space-time scale difference between cavities and model grid. Furthermore, the lack of adequate constraint data for this scale dictates a more simplified approach to minimize the number of tunable parameters.

Only a few subglacial hydrology models have been described in the literature for continental-scale ice sheets. Of these models, some of the more advanced include the models developed by Flowers (2000, 2008), Johnson (2002), Arnold and Sharp (2002), Goeller et al. (2013), Bueler and van Pelt (2015), and Gudlaugsson et al. (2017). These models take various approaches to simulate the flow of basal water using physically-based equations.

The original work of Flowers (2000) developed a physics-based, multi-component model that included englacial, subglacial, and groundwater (aquifer) hydrology. The subglacial component of this model simulated the flow of water as a distributed system via Darcian flux. The equations were cast in a finite-volume discretization (Patankar, 1980) and advanced in time using an iterative Newton-Krylov method on a 40x40 metre grid. Later work on that model included a channelized flow that coexisted alongside the distributed system and allowed exchange between the two systems (Flowers, 2008).

Johnson (2002) developed a continental-scale model with a 5 km grid resolution. In this model, the water is transported underneath the ice sheet via a tunnel (channelized) system solved using the turbulent Manning pipe flow equation. The aquifer in this model was simply a parameter that drained a percentage of the available water in the grid cell. The equations of Johnson (2002) were solved using the Galerkin method for finite-element discretization.

The work of Arnold and Sharp (2002) attempts to model the flow of water with both distributed and channelized systems. The model determines the type of system operating in each grid cell based on the water flux in the grid cell. When the flux allows the grid cell to exceed the "orifice stability parameter" (Kamb, 1987), then the grid cell has a channelized system, otherwise, it is a distributed system. The model integrates the basal water fluxes down the hydraulic potential. From the fluxes, the model determines the drainage system present, which is a different method employed from those used in the previous two models and the one developed herein.

Goeller et al. (2013) considers a distributed system that covers the base of the ice sheet. As a simplification, the basal water pressure is assumed to be approximately equal to the ice overburden pressure. This simplifies the hydraulic gradient to follow the bed and ice geometries. Water flux out of a grid cell is limited in the case that it would lead to negative water by applying a multiplier to the out-fluxes that lower their values to the desired limit. Their model does not consider any channelized system. A similar model was used in Gudlaugsson et al. (2017) applied to the Eurasian ice sheets that covered Northern Europe and parts of Asia during the last ice age.

Carter et al. (2017) created a 1D-model for the simulation of lake drainage beneath Antarctica. Their model did a detailed comparison to the rate and frequency of water drainage from a subglacial lake via R-Channels and canals cut into the underlying sediment. Their model showed that the canal drainage system provided better estimations for cold ice, whereas R-Channels

would be more common in warm, temperate ice, such as near the terminus of Greenland that is also fed by surface run-off. Similar conclusions were drawn from Dow et al. (2015).

The basal hydrology model described here combines features from the above models to create a relatively fast subglacial hydrology model for continental-scale contexts. Following the work of Arnold and Sharp (2002), the basal drainage system is allowed to have both distributed and channelized drainage systems with a condition for determining which basal system is present. While conceptually similar, the implementation is rather different. In this model, the drainage system is initially assumed to be distributed, as in Flowers (2000), with basal fluxes computed under the same Darcy flow approximation. The distributed system in a grid cell is switched to a channelized drainage system when the flux exceeds a critical value developed in Schoof (2010). The switching condition explained in Schoof (2010) is for a R-Channel, but the model can allow for other conditions to be used that better suit other channelized drainage types. Starting at the grid cells that meet the switching condition, channelized systems are created by following the path of steepest hydraulic gradient until a potential well or exit is reached. R-Channel drainage is imposed instantaneously. For developmental expediency, the aquifer physics of Flowers (2000) are replaced with the drainage parameter from Johnson (2002).

Bueler and van Pelt (2015) developed a subglacial hydrology model for the PISM model. Similar to BraHMs, their hydrology model simulates the subglacial water flow using a Darcian flux and limit the basal pressure to the ice overburden pressure (due to long time scales). Their model consists of several basal components, including a water-filled till layer and an effective cavity-based water storage. The model presented by Bueler and van Pelt (2015) does not have any channelized flow mechanisms, which is a major source of water flow/drainage beneath ice sheets, as discussed in Section 2. It is unclear as to how well this model compares to BrAHMs in terms of speed due to the vast difference in model grid and computer usage. The model of Bueler and van Pelt (2015) incorporates the opening and closure of cavities which is necessary for high resolution modelling of present-day glaciers and ice sheets, but can be replaced by low-resolution physics for longer time scales and larger spatial scales, where data is sparse. The incorporation of cavity opening and closure would require computation resources that may be prohibitive for long-term, continental-scale models.

Calov et al. (2018) uses SICOPOLIS in their study on the future sea level contributions of the Greenland ice sheet. The basal hydrology model used in SICOPOLIS is for large-scale grid cells like BrAHMS. The model assumes a thin film of water, resulting in zero effective pressure (meaning the hydraulic potential is simply related to bed elevation and ice sheet thickness). The model first determines a down gradient path from each grid cell to the ocean/boundary. Any depressions are filled with water (akin to lakes) and is given a small gradient so the down gradient solver can continue. From this the water level can be calculated based on the input of meltwater and the hydraulic gradient. This is a rather different approach than BrAHMs as BraHMs attempts to model the physical evolution of the water, allowing for varied flow of water, non-zero effective pressures, and the time-evolution of lake growth under Darcy flow.

Hoffman and Price (2014) also developed a physically-based model to be used as a part of CISM. This model is rather detailed in combining cavity formation (providing water storage) and a method to form Röthlisberger channelized flow. Analysis of this model looked at fine (100 m x 100m) grids and for shorter periods of time (on the order of days). While they do not mention the speed of their model, given that they model the growth and decay of channels, it is unlikely that the model would be

suitable for longer time scales as the time stepping must be small to capture the transient nature of channelized flow. Likewise, for larger grid sizes, the effects of distributed systems (cavities, thin films, etc..) can be averaged out, saving on computation with minimal lost of generality in the results.

A distinguishing feature of this model is the numerical time stepping scheme. The model uses a combination of Heun's method and the leapfrog-trapezoidal schemes, which are iterative predictor-corrector schemes. The latter scheme has been used in the more demanding case of ocean modelling (Shchepetkin and McWilliams, 2005). The combination of these two methods (see appendix A2) proves to be robust and stable with quick convergence to the final solution.

The hydrological model has been incorporated into the Glacial System Model (Tarasov and Peltier, 1999; Tarasov et al., 2012). Below, we further detail and validate the subglacial model. We document water pressure and thickness sensitivity to hydrological parameters. Example results for the past North American ice complex are presented. Conclusions are summarized in Section 7.

## 2 Subglacial Drainage Systems

Subglacial drainage systems can be characterized as belonging to one of two categories.

### 2.1 Distributed Drainage System

There are several ways that water can be distributed underneath the ice: Water can be stored via a thin film (Weertman, 1972) between the bed and the ice; water can be stored on the lee side of bed protrusions to form a linked-cavity system (Kamb et al., 1985); braided canals (Clark and Walder, 1994) are formed as water cuts into underlying sediment; and water can flow through a porous medium via Darcian flow (Flowers, 2000). Distributed systems are inefficient at draining water. These types of systems, therefore, lead to a build-up of basal water pressure under the ice sheet.

### 2.2 Channelized Drainage Systems

The channelized drainage system is, to a certain degree, the obverse of the distributed drainage system. This system has a lot of water concentrated in a small area of the glacial bed and transports water quickly. Since channelized systems are efficient at draining water, they tend to decrease the water pressure, thereby increasing basal friction between the ice and the bed. Thus, channelized systems are associated with slow flowing ice regimes and are often seasonal. Bartholomew et al. (2011) provides evidence that sliding velocities near the margins of the Greenland Ice Sheet are lower in the late summer than earlier in the summer, likely as an indication of a switch from a distributed to a channelized drainage system. There are two types of channelized drainage systems: Nye Channels that are incised down into the substrate (Walder and Hallet, 1979), and R-channels that are incised up into the ice (Röthlisberger, 1972).

## 3 Glacial System Model

For the analyses presented herein, the subglacial hydrology model is passively coupled to the Glacial Systems Model (GSM). Full two-way coupling was turned off to isolate the dynamical response of the basal hydrology model. The GSM is composed of a thermo-mechanically coupled ice sheet model (using the shallow ice approximation), locally 1D diffusive permafrost resolving bed thermal model (Tarasov and Peltier, 2007), fully coupled diagnostic surface drainage and lake storage module (Tarasov and Peltier, 2006), visco-elastic bedrock response, positive degree-day surface mass-balance with refreezing, and both marine and lacustrine calving parameterizations (Tarasov and Peltier, 1999, 2002, 2004; Tarasov et al., 2012).

The evolving temperature field ($T$) of the ice sheet is determined from conservation of energy:

$$\rho_i c_i(T) \frac{\partial T}{\partial t} = \frac{\partial}{\partial z}\left\{ K_T(T) \frac{\partial T}{\partial z} \right\} - \rho_i c_i(T) \boldsymbol{u} \cdot \boldsymbol{\nabla} T + E_d \tag{1}$$

with $c_i$ representing the specific heat of ice, $\rho_i$ is the density of ice (910 kg/m$^3$), $K_T$ is the thermal conductivity of ice, $\boldsymbol{u}$ is the sliding velocity of the ice, and $E_d$ is the heat created from the deformation of ice. As is standard, the horizontal diffusion component is ignored given the scales involved. The fully coupled ice and bed thermodynamics are solved via an implicit finite volume discretization in the vertical and explicitly for the horizontal advection component of the ice thermodynamics. Basal temperature is limited to a maximum of the pressure melting point, with excess heat used to melt basal ice.

Basal sliding uses Weertman type sliding relations (i.e., function of driving stress) with power law 3 for hard bed and power law 1 for soft bed with sliding onset linearly ramped up starting from $0.2^o\,C$ below the pressure melting point.

## 4 Subglacial Hydrology Model

### 4.1 Model Description

For brevity and clarity, this section discusses the physical and numerical concepts of the hydrology model developed in Kavanagh (2012). The appendix provides details on the spatial discretization of the equations and the time stepping using the Heun/Leapfrog-trapezoidal scheme. This scheme is second order accurate. The model dynamically adjusts its internal timestep to ensure the CFL criteria is satisfied (with timestep set to $F_{CFL}\times$ minimum CFL timestep). Both of these features contribute to the stability of the model.

The dynamical evolution of distributed drainage is extracted from the mass continuity equation. Written in conservative form, the equation is

$$\frac{\partial w}{\partial t} + \boldsymbol{\nabla} \cdot \boldsymbol{Q} = \dot{b} + d_{s:a} \tag{2}$$

with $w$ being the water thickness.

$\dot{b}$ is the meltwater source from the ice (negative if water refreezes to the ice), For the scope of this initial investigation, we assume no transmission of ice surface melt to the base. Observationally this is known to be false (Zwally et al., 2002), but the dependence on ice thickness, ice temperature profile, and ice strain profile makes this an issue deserving of its own focused study. The other source term, $d_{s:a}$, represents the drainage into the underlying aquifer.

The water flux, $\boldsymbol{Q}$, is given by Darcy's law:

$$\boldsymbol{Q} = -\frac{Kw}{\rho_w g}\boldsymbol{\nabla}\{P + \rho_w g z_b\} \tag{3}$$

where $K$ is the hydraulic conductivity of the underlying till, $\rho_w$ is the density of water ($1000\,\text{kg/m}^3$), $g$ is the acceleration due to gravity ($9.81\,\text{m/s}^2$), $z_b$ is the topographical bed elevation, and $P$ is the water pressure beneath the ice.

The distributed flow of water beneath the ice sheet can come in many forms (such as cavities, Nye-channels, thin film, and flow through porous sediment). It is unclear however, the extent to which the details of these flow mechanisms matter under large spatial scale separation and mechanistic heterogeneity. We therefore use the large difference in scale between glacial cycle ice sheet model grid cells (O(10km)) and distributed sub-glacial flow structures (O(10 m) or less) to justify the choice of the diffusive Darcy flow equation for BrAHMs.

We use an empirical relation for water pressure from (Flowers, 2000, page 68) given by

$$P = P_I \min\left[\left(\frac{w}{h_c}\right)^{7/2}, 1\right] \tag{4}$$

where $P_I$ is the ice overburden pressure. $P$ is limited to ice overburden pressure when $w \geq h_c$. Saturated sediment water thickness, $h_c$, equals till thickness times porosity and is effectively the water thickness that the till can hold before becoming over-saturated (at which point the excess water is stored between the till and the ice). Flowers (2000) derived this equation by

considering sub-grid variation in bed elevation and associated sediment thickness (and therefore water thickness, all for the context of 40 m x 40 m grid-cell modelling of Trapridge Glacier). A further consideration (without the overburden limit) was that the nonlinearity would address dynamic adjustments in porosity and prevent stiffness in the dynamic equations caused by unrealistic heterogeneity in the modelled water distribution. Though derived for glacier-scale flow through a heterogenous macroporous sediment layer, our working hypothesis is that this empirical relation approximately captures large scale pressure

response for subglacial distributed flow through any heterogenous structure (be it a mix of cavities of different size, patchy sediment, Nye channels...). The limiting of water pressure to overburden is justified by the low likelihood of water pressure exceeding the overburden pressure for any significant amount of time on glacial cycle timestep scales.

The channelized system is likened to a system of R-Channels (tunnels incised upward into the ice). Numerically, the model first calculates the water flux from the Darcian flow (equation 3). Channelized flow is invoked when that flux exceeds a critical

value for the stability of the distributed regime given in Schoof (2010) as

$$|\boldsymbol{Q}| < \frac{|\boldsymbol{u}_b| Z_h}{(\rho_i L)^{-1}(\alpha - 1)\boldsymbol{\nabla}(P + \rho_w g z)} \tag{5}$$

where $u_b$ is the basal sliding velocity of ice, $Z_h$ is the bedrock protrusion height, $L$ is the latent heat of fusion of ice, and $\alpha = 5/4$.

To simulate the change between different drainage systems, at regular user-defined intervals, the channel flow subroutine is

called. Grid cells for which the water flux exceeds the distributed flow stability criterion (equation 5) are marked as tunnel grid cells. A down hydraulic gradient solver (Tarasov and Peltier, 2006) instantaneously[1] moves water down the path of steepest

---

[1] During the tunnel flow, no model time is stepped as tunnel drainage is computed diagnostically.

potential gradient (channelizing grid cells along that path) until there is no grid cell with a lower hydraulic potential (so forms subglacial lake) or the water exits the ice sheet. The solver considers all adjacent grid cells (including corner adjacency) when searching for the lowest hydraulic potential. Once the tunnel water transport is complete, the tunnels are assumed closed and the distributed flow algorithm continues.

## 4.2   Model coupling

BrAHMs is highly modular and designed for asynchronous coupling at user specified timesteps. Aside from basic grid infor-
mation, for each call, the hydrology model requires the following input fields: ice thickness, basal elevation, sea-level, basal ice temperature, basal melt rate, and basal sliding velocity of the ice. For two-way coupling, the relevant outputs from BrAHMS are basal water pressure and thickness.

Given that there is no lower limit to coupling timesteps, synchronous coupling can also be implemented. For two-way coupling, sensitivity tests are recommended to determine the appropriate coupling timestep for the relevant context.

# 5   Model Validation

## 5.1   Setup

The basal hydrology model was subject to several validation tests with synthetic axisymmetric ice sheets. The three setups are:
symmetrical ice sheet on a flat bed, symmetrical ice sheet on a dilating (sinusoidally-wavy) bed, and a symmetrical ice sheet on an inclined plane.

### 5.1.1   Ice Sheet Profile

The continental-scale ice sheet model used in these tests has a profile that follows a normal distribution from the centre of the ice sheet to the terminus and is symmetric around the centre (i.e., bell-shaped ice sheet), according to the equation

$$H(d) = (H_{max} - H_{min}) * \exp\left[-\left(\frac{d}{\sqrt{2}H_d r}\right)^2\right] + H_{min} \qquad \text{For} \quad (d < r_t) \tag{6}$$

where $H_{max}$ is the ice thickness at the ice divide (the centre), $H_{min}$ is the thickness at the terminus, $H_d$ is a normalized (by radius) standard deviation that defines how the profile spreads out, $r_t$ is the distance to the terminus, and $d$ is the distance from the ice divide, given by

$$d(\theta, \phi) = \cos^{-1}\left[\sin(\theta_c)\sin(\theta) + \cos(\theta_c)\cos(\theta)\cos(\phi - \phi_c)\right] R_e \tag{7}$$

where $\theta$ and $\phi$ are the latitude and longitude, and $R_e$ is the radius of the Earth (all parameter values are listed in Table1).

### 5.1.2 Incline and Dilating Bed Profiles

The bed for the inclined plane is given by

$$d_N = \cos^{-1}[sin(\theta_N)\sin(\theta) + \cos(\theta_N)\cos(\theta)]Re$$
$$d_c = \cos^{-1}[sin(\theta_N)\sin(\theta_c) + \cos(\theta_N)\cos(\theta_c)]Re$$
$$z = Z_{max} - \frac{Z_{max}d_N}{d_c} \tag{8}$$

to test the flow of water, where $Z_{max}$ is the maximum elevation and the slope is calculated such that $z = 0$ at the south

$(d_N = d_c)$.

The dilating bed given by

$$z(\theta, \phi) = min\left[(Z_{min})\cos\left(\frac{(\theta - \theta_C)R_e}{5}\right)\cos\left(\frac{(\phi - \phi_C)R_e}{5}\right), 0\right] \tag{9}$$

where $Z_{min}$ m is the maximum depth of the bed (all parameter values are listed in Table 1).

### 5.2 Model Runs

It should be noted that the model is based on spherical polar coordinates (as it is designed for modelling large sections of the Earth's surface), and so the figures presented here are akin to the Mercator projection [2] (mapping polar coordinates to Cartesian).

Tables 1 and 2, at the end of this section, list the parameters used in the validation analysis. For Table 2, the values used for the validation tests are listed in the "Value" column.

In the model runs, the ice sheets starts from the ground (at $t_{now} = 0$) and grows until 50% of the model run-time ($t_{half}$). The ice thickness grows according to eqn. 6 multiplied by the ratio $t_{now}/t_{half}$. When $t_{now}$ is greater than $t_{half}$, the ice sheet is at its maximum size (as shown in fig. 1a).

To facilitate the growth of the subglacial hydraulic system, a constant melting at the base of the ice is applied in a 'ring' of uniform thickness near the terminus, with 0.6 m/yr of melting at the terminus and decreasing linearly to 0.4 m/yr at the inside

of the 'ring'. However, if there is no ice where the ring of meltwater is defined, then the value of melt, $M_d$, is set to zero until there is ice, in which case it would take the value defined by the equation

$$M_d(d) = M_t - (M_t - M_i)\left(\frac{r_t - d}{c_r}\right) \qquad \text{For} \quad (r_t - c_r < d < r_t) \tag{10}$$

where $M_t$ is the melt rate at the terminus, $M_i$ is the melt rate on the inside of the melt ring, and $c_r$ is the thickness of the ring from the terminus into the innermost melting point.

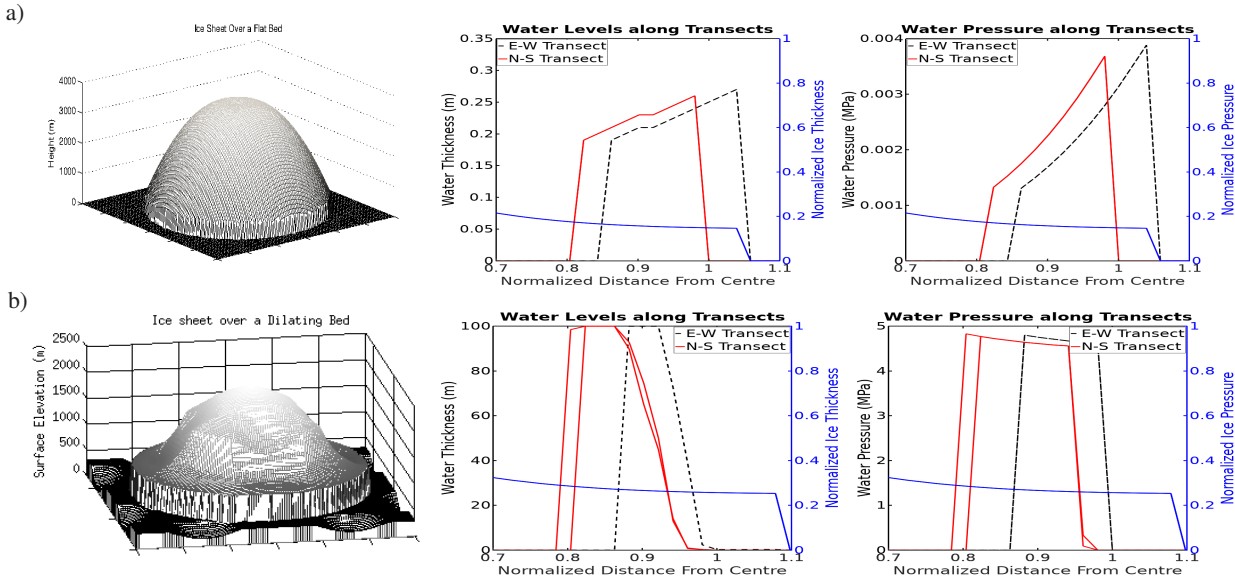

**Figure 1.** Simple synthetic ice sheet testing scenarios. Transects are plotted as the absolute, normalized distance from the centre. a) shows a dome-shaped ice sheet placed on a flat bed resulting in symmetric results. The results in b) show the ice dome on a dilating bed. Lake formation occurs where the ice sheet is relatively flat and the topography dips.

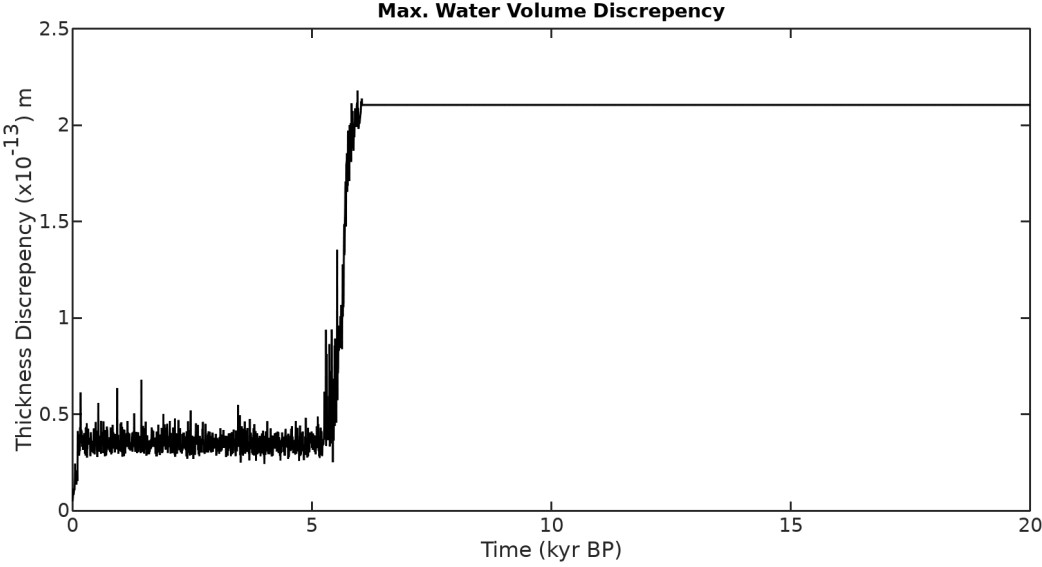

**Figure 2.** Maximum mass balance discrepancy over time.

### 5.3 Validation Results

#### 5.3.1 Symmetric Ice Sheet on Flat Bed

The first setup tested was for the ice sheet on a flat bed. For the bell-shaped ice sheet on a flat surface (see Figure 1a), the model is mass conserving on the order of $10^{-12}$ m of water thickness within a grid cell (Figure 2). For this case, the water drains radially away from the ice sheet under the influence of the basal water pressure. The average water thickness was $0.0801 \pm 0.1357$ m for the grid cells that contained water.

The transects in Figure 1a show how the water thickness/pressure profiles change along two cross-sections of the sheet through the centre (Transects are plotted along the absolute-value, normalized distance from the centre of the ice sheet. For example, The water profile north of the centre is plotted on top of the south water profile for comparison). For the case of the flat bed, both transects appear to be single lines, showing that there is a N-S symmetry and and a E-W symmetry. Both transects are slightly offset from each other due to the grid discretization, but otherwise look similar.

#### 5.3.2 Symmetric Ice Sheet on Dilating Bed

The next test was designed to show the formation of lakes by the model. The ice sheet for this study, as seen in fig. 1b, is similar to the other ice sheets, but has its centre height ($H_{max}$) lowered from 3500 m to 2000 m to smooth out the ice sheet toward the terminus, as the previous profile was too steep - leading to no lake formations.

Figure 1b shows that (in areas where the ice is relatively flat and there is a dip in the bed) the hydrology model does allow for the build-up of water into subglacial lakes, reaching up to 100 m of water thickness in places.

The transect plots show that there is a slight asymmetry that arises in the results (there appears to be two red curves). Under perfect symmetries, the tunnel solver will break symmetry in its down-slope search algorithm. While the results are not shown here for brevity, when the tunnel solver is turned off, the results do not show any discernible asymmetry. The asymmetry due to the inclusion of the tunnel solver is unlikely to be an issue in more realistic cases where the ice sheet would lack such symmetry.

#### 5.3.3 Symmetric Ice Sheet on an Inclined Plane

Figure 3 indicates that there is an asymmetry of N-S transects (in the interior of the ice around 0.8 normalized distance), that does not occur in the E-W transect which still maintains its symmetry as in the flat bed case. This is due to the slope of the bed that results in a build up of water in the northern section of the ice sheet that reduces the hydraulic gradient in comparison to the hydraulic gradient in the southern section. The average water thickness in this scenario is $0.0802 \pm 0.1357$ m.

Figure 3 shows the results for both the normal and double resolution grids. These two results are nearly identical, except the basal water is slightly thicker in the higher resolution plot ($0.0805 \pm 0.1365$ m) and are slightly offset from each other due to the changes in grid resolution. This suggests that the model is convergent at finer grid resolutions.

---

[2]set between 17–40°N for the simplicity of avoiding any issues such as having negative values for generating the synthetic geometry.

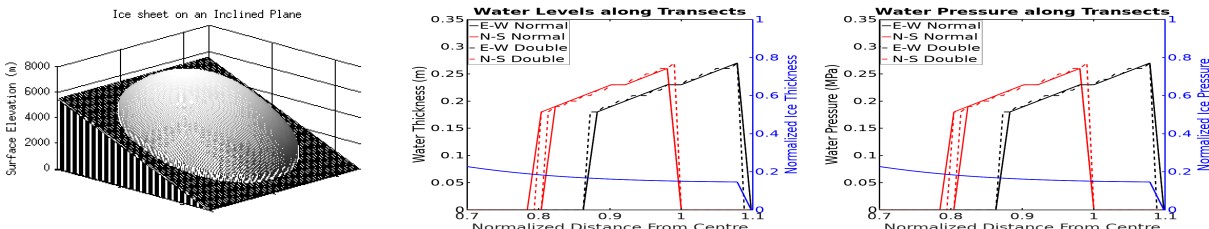

**Figure 3.** Plots of the bell-shaped ice sheet on an inclined plane. This series also tests the convergence of the model. "Normal" refers to the model using the same resolution as the previous tests and "Double" refers to the tests that has twice the resolution (finer) grid than the other tests.

| Parameter | Flat Bed Test | Incline Test | Dilating Bed Test |
|:---:|:---:|:---:|:---:|
| $\phi_c$ | | -90° | |
| $\theta_c$ | | 17.71° | |
| $\theta_N$ | | 40° | |
| $H_{max}$ | 3500 m | | 2000 m |
| $H_{min}$ | | 500 m | |
| $H_d$ | | 0.3 m | |
| $r_t$ | | 1592.8 km | |
| $R_e$ | | 6371 km | |
| $M_t$ | | 0.6 m/yr | |
| $M_i$ | | 0.4 m/yr | |
| $c_r$ | | 318.55 km | |
| $Z_{min}$ | 0 m | | -300 m |
| $Z_{max}$ | 0 m | 6000 m | 0 m |

**Table 1.** Model parameters used in validation studies.

### 5.3.4 Model Stability Test

Lastly, the model was tested for stability by shocking the system with sudden changes in the meltwater production. For this test, the base case of the ice dome lying on a flat bed (the same scenario as Figure 1a), only the meltwater production from equation 10 was modified by a time-dependent multiplier

$$M_d^*(d,t') = \xi(t') \times M_d(d)$$

$$\xi(t') = 1 + \sin\left[2\pi\left(\frac{t'-1/8}{80}\right)\right]\left(\Delta[t'-\frac{1}{8}]-\Delta[t'-\frac{1}{4}]\right) + \sin\left[2\pi\left(\frac{t'-3/8}{800}\right)\right]\left(\Delta[t'-\frac{3}{8}]-\Delta[t'-\frac{1}{2}]\right)$$

$$+ \sin\left[2\pi\left(\frac{t'-5/8}{80}\right)\right]\left(\Delta[t'-\frac{5}{8}]-\Delta[t'-\frac{3}{4}]\right) + \sin\left[2\pi\left(\frac{t'-7/8}{800}\right)\right]\left(\Delta[t'-\frac{7}{8}]\right) \tag{11}$$

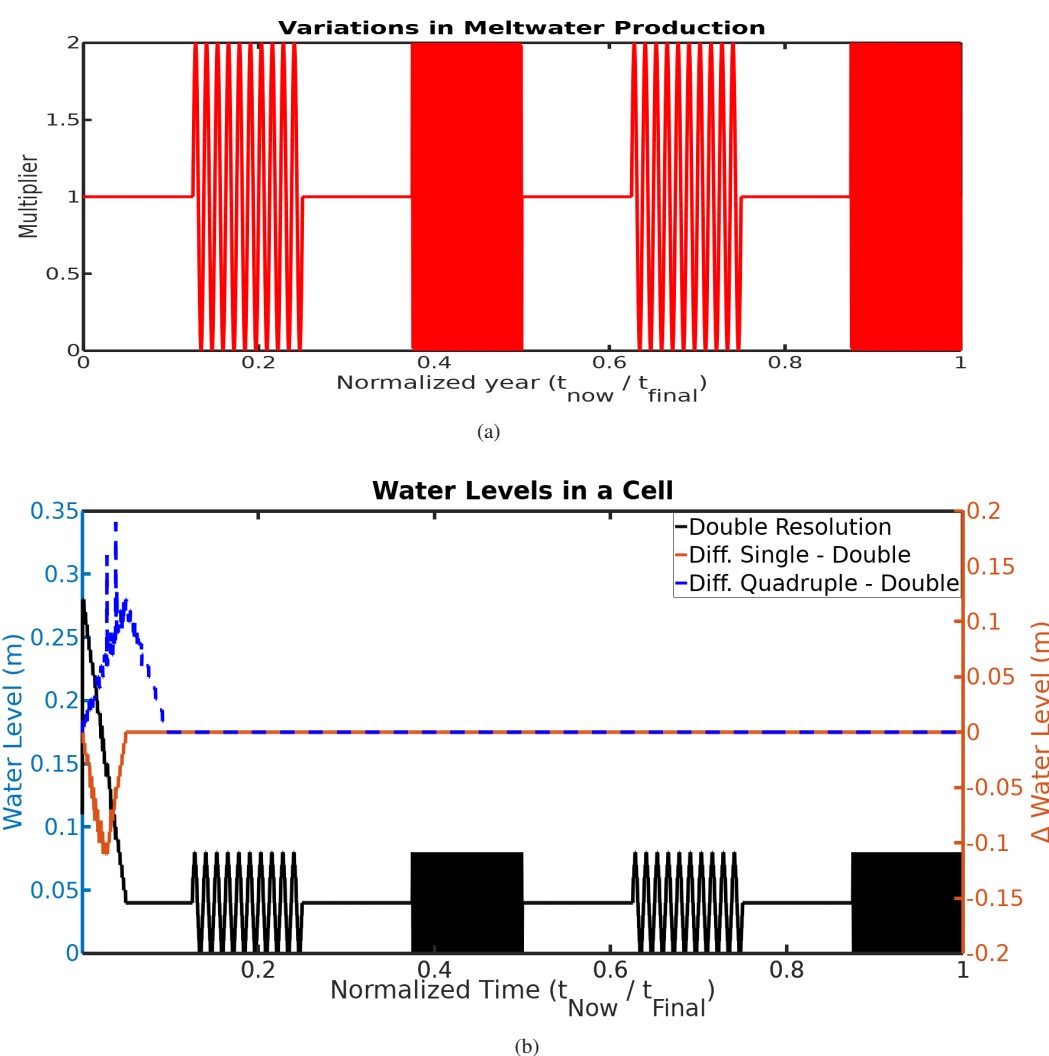

**Figure 4.** Time series analysis of basal water level response to time-varying meltwater production, showing that the dynamic time-stepping of the model is capable of providing stable, realistic results under sudden changes.

| | List of Model Parameters | | | |
|---|---|---|---|---|
| | Represents | Value | Range | Reference |
| $F_{CFL}$ | Prevents breaking CFL | 0.50 | 0.1–0.9 | N/A |
| $dt_{max}$ | Maximum allowable time step | 1/12 yr | 1/36–1 yr | N/A |
| $D_r$ | Percent of water drained to aquifer | 2.00% | 0–0.07 | Johnson (2002) |
| $dt_{tun}$ | Time interval between tunnel checks | 1/4 yr | 1/12–1 yr | N/A |
| $h_c$ | Saturated sediment water thickness | 1.00 m | 0.1–2 m | Person et al. (2012) |
| $k_a$ | Steepness of conductivity transition | 15 | 5–60 | Flowers et al. (2005) |
| $k_b$ | Affects when conductivity transitions | 0.65 | 0.25–0.95 | Flowers et al. (2005) |
| $K_{min}$ | Minimum hydraulic conductivity | $10^{-7}\frac{m}{s}$ | $10^{-9}$–$10^{-5}$ | Flowers et al. (2005) |
| $K_{max}$ | Maximum hydraulic conductivity | $10^{-5}\frac{m}{s}$ | $10^{-7}$–$10^{-3}$ | Flowers et al. (2005) |
| $Q_{sc}$ | Tunnel formation condition multiplier | 1.00 | $10^{-4}$–$10^{4}$ | N/A |
| $T_c$ | Basal freezing temperature below PMP | $-2.00°C$ | $-3$–$-0.5°C$ | N/A |
| $Z_h$ | Bedrock bump height | 0.10 m | 0.01–0.5 m | Kamb (1987) |

**Table 2.** Chosen values for the baseline model run for synthetic and North American test runs.

where $t' = t_{now}/t_{final}$ is the normalized time ($t_{final} = 800$ years), and $\Delta(t')$ is the Heaviside step function. $\xi(t')$ is plotted in Figure 4a. Each impulse lasts for 1/8-th (100 years) of the model run. The lower frequency impulses contain 10 wavelengths each (period of 10 years), and the higher frequency impulses contain 100 complete wavelengths (period of 1 year). Each impulse will vary the meltwater production from a range of 0–2 times the base value.

The Figure 4b show how BrAHMs is affected by the varying meltwater production for changes in the CFL condition affecting time step size and to changing the resolution of the grid. The analysis showed that there was no discernible difference from halving the CFL condition, so the results of changing the CFL condition were omitted. While there are some differences (along this particular transect) related to changes of the grid resolution, none of the plots exhibit any unrealistic behaviour due to the cycling of the meltwater production.

## 6   Model Results Coupled to the GSM

The North American ice sheet model used herein is from a large ensemble Bayesian calibration as detailed in Tarasov et al (2012). Model runs start from 122 ka under ice free conditions.

### 6.1   The Model Parameter Set

Due to the complex nature of basal hydrology and the spatial and temporal scales for the current context, there are many processes that are approximated through parameterizations. As such, there are a number of poorly constrained parameters in the hydraulic model (these are listed in Table 2).

The first parameter, $F_{CFL}$, is used to control the time-stepping of the model. As the model runs an explicit time scheme, it is subject to the CFL condition for stability. To help prevent the model from breaking the CFL condition, the model time step is dynamically altered to prevent the maximum basal water velocity from exceeding the CFL velocity. $F_{CFL}$ determines the maximum allowable basal water velocity as a fraction of the CFL velocity. Should a time-step exceed the CFL condition (potentially leading to instabilities), the last time-step is redone with a smaller $\Delta t$ such that the CFL condition is not broken.

The simplified aquifer drainage of Johnson (2002), uses an aquifer that simply drains a percentage of the present water in a grid cell. The percentage of water drained in this model is represented by the $D_r$ parameter ($d_{s:a} = D_r w$).

Due to the small time steps (relative to glacial modelling) involved in the basal hydrology model, it would become computationally expensive to check for tunnels at each time step. As such, $dt_{tun}$ determines the frequency at which the model checks for the formation of channelized flow.

For clarity of this initial analysis, results presented herein are with a uniform basal sediment cover over the whole bed for the duration of the run. The sediment cover was specified by $h_c$.

The water flux between grid cells is directly proportional to the hydraulic conductivity of the sediment. For each run, the conductivity was allowed to vary between a minimum and maximum value defined in the range of $K_m$. The hydraulic conductivity is allowed to vary to account for the fact that the till will expand as it is filled with water, allowing for the water to flow with less resistance. The transition between low and high conductivity is controlled by the parameters $k_a$ and $k_b$ given by the equation (Flowers, 2000)

$$\log(K) = \frac{1}{\pi}(\log[\frac{K_{max}}{K_{min}}])\tan^{-1}\left[k_a\left(\frac{w}{h_c} - k_b\right)\right] + \frac{1}{2}(\log[K_{max}K_{min}]) \qquad (12)$$

which is a constitutive equation of the logarithmic form of the upstream area of a grid cell (Flowers, 2000, page 80). Data from Trapridge Glacier shows a similar relation between upstream area and water pressure as the hydraulic conductivity equation (a high and low regime with a transition zone). Flowers (2000) assumes that the upstream area is related to the connectivity in the grid cell (the more connected a grid cell is, the more upstream area it should have). This would suggest that the hydraulic conductivity (its connectivity) is dependent on the water level, and is of the form of equation 12.

In equation 12, $k_b$ affects, as a fraction of $h_c$, when the transition between begins ($k_b/h_c$ is the halfway point of the transition). $k_a$ affects the slope of the transition curve. For larger values of $k_a$ the transition becomes sharper, leading to quicker transitions. Lower values of $k_a$ lead to slower transitions with more intermediate values for the conductivity between the two extremes (See Figure A1). Equation 12 is meant to capture the dependence of hydraulic conductivity on the pore size of the till, which is related to the amount of water in the till.

As the base of the ice sheet becomes colder, the ice should begin to freeze to the bed, preventing water from flowing there. Due to the 40 km resolution of the grid, it is unlikely that the entire bed in a grid cell would be frozen completely when the grid cell basal temperature crosses the pressure melting point. Water could therefore potentially flow through a frozen grid cell (in the unfrozen places), but the water should have a harder time as it has fewer pathways to flow across. In the hydrology model, this is represented by parameter $T_c$, which acts to reduce the conductivity as a function of temperature. When the basal temperature is close to the pressure melting point (PMP), there is little change in the hydraulic conductivity. Conductivity

decreases to an extremal low value as the temperature approaches the value of $T_c$. In the model simulations, the value of $T_c$, relative to PMP, is tested from $-0.5°C$ to $-3.0°C$. As a simplifying assumption, the hydraulic conductivity of a frozen grid is set to $10^{-14} \frac{m}{s}$, but can be easily modified to follow a temperature-dependent profile to capture sub-grid variation in basal temperatures.

Tunnel formation has a direct impact on basal water pressure. To further test this impact, an enhancement factor, $Q_{sc}$, was introduced to equation 5 as a multiplier to the condition for tunnel flow. Higher values of $Q_{sc}$ will increase the switching condition, leading to less tunnel formation, whereas lower $Q_{sc}$ will increase the amount of tunnel formation.

## 6.2 The Baseline Model

Our choice of baseline model for the sensitivity analysis was solely based on mid-range values for parameter uncertainty ranges

and not on any sort of tuning. As such, results presented here have an exploratory instead of predictive focus. Basal hydrology fields for the baseline model near last glacial maximum (LGM) are shown in Figure 5. There is a greater extent and generally thicker basal water at 22 ka than at 18 ka. Regions of low basal effective pressure (defined as ice overburden pressure minus basal water pressure) in the model are generally associated with ice streaming. As the results are for a model configured with no basal drag dependence on basal water, this correlation is potentially due to two factors. First, high basal velocities will

increase basal heating and thereby basal meltwater production. Second, the basal flux trigger threshold for initiation of tunnel drainage (equation 5) is proportional to basal velocity.

    As the water is removed from 22 ka to 18 ka, some of the areas experience a large increase in basal effective pressure. To account for dependence on baseline amounts of basal water, our sensitivity tests consider both the 22 and 18 ka time-slices. Figure 6a shows model sensitivity at 22 ka when the baseline model total water volume is higher. The most important

parameter is the aquifer drainage parameter, which is the proportion of water drained locally out of the system. This simplified parameterization of the aquifer can quickly drain a lot of water as it does not have to flow to the terminus to escape and does not return it to the ice-bed interface. In Figure 6b, at 18 ka, the aquifer drainage is still the most important parameter, but its impact is less noticeable since there is less water to drain away from the bed.

    The sediment thickness parameter ($h_c$) shows a 28% drop in water volume over the range of values at 22 ka. At 18 ka the

impacts of $h_c$ are greatly reduced and has no effect on water volume when raised above the baseline value. This is due to the nonlinear relation between water pressure and the sediment thickness from equation 4. In areas where the water level is only a small fraction of the sediment thickness, the basal water pressure will be practically zero. At 18 ka, when the water level is low, an increase in sediment will have little effect on the results.

    The runs with the basal freezing value closer to the PMP have about a 12% increase in basal water volume, as expected

due to the increased likelihood of ice frozen to the bed hindering the flow of water. In comparison to other parameter results, varying the value of the basal freezing parameter, $T_c$, does not alter the water storage significantly. This is expected as regions where the basal temperature is below the PMP have no subglacial meltwater production.

    The tunnel criterion scaling factor $Q_{sc}$, show almost no impact in times of high water storage, but shows a drop of up to 80% in water volume at 18 ka. During this time, the model is sensitive to $Q_{sc}$ because the lower water levels are less likely to

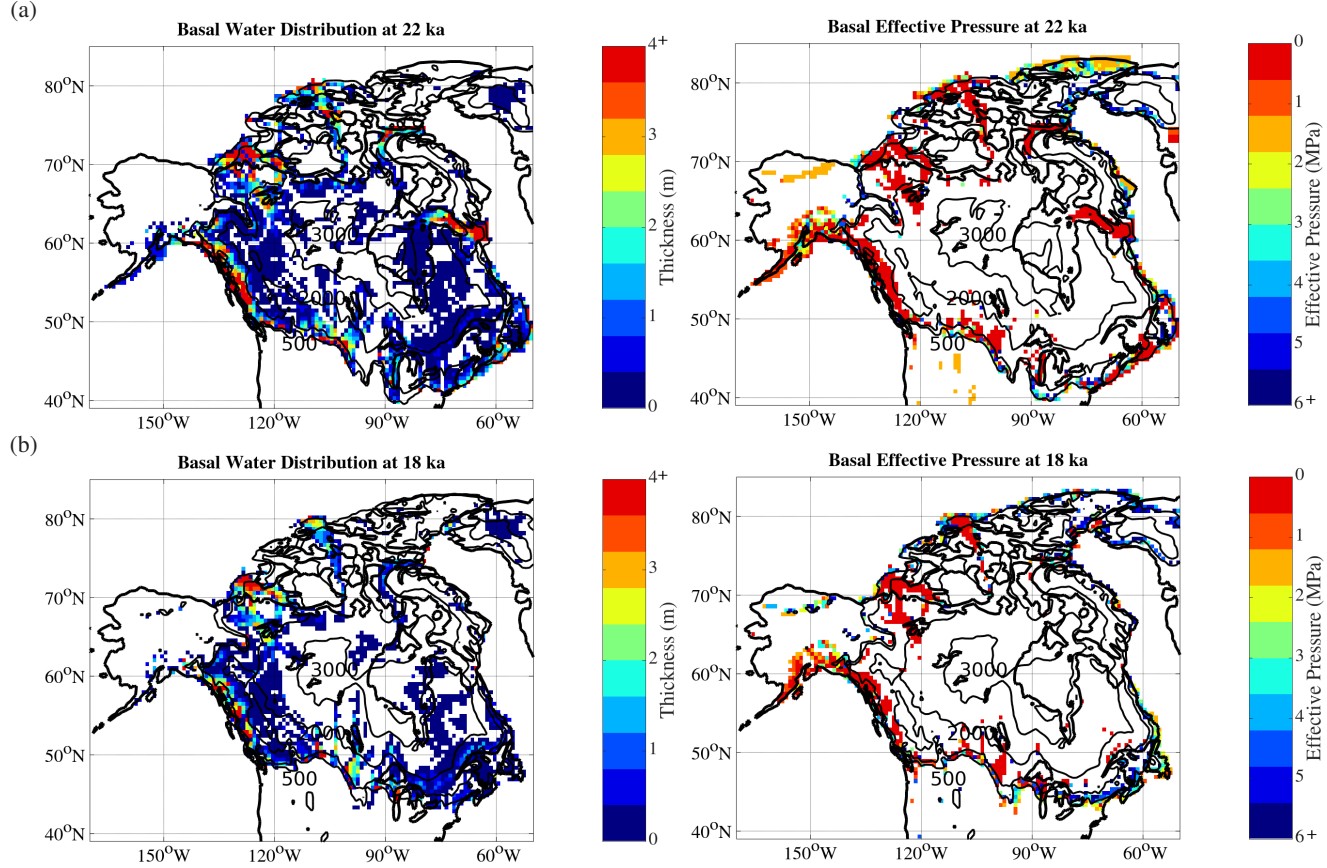

**Figure 5.** Basal water profiles for a) 22 ka when the total water volume is high (mean thickness: $1.0644 \pm 2.8317$ m, max thickness: $86.40$ m), and b) 18 ka, after a large reduction in total basal water volume (mean thickness: $0.6918 \pm 1.3160$ m, max thickness: $24.91$ m). 500 m contours intervals for surface elevation are also shown.

form tunnels than the thicker values at 22 ka. Lowering $Q_{sc}$ allows more tunnels to form, which drains the water, keeping the water volume down.

The bedrock bump height, $Z_h$, has a similar effect to $Q_{sc}$ since it affects tunnel formation as well. Larger values of $Z_h$ allow the cavity system to retain more water before filling up and becoming unstable. This allows the runs with higher $Z_h$ to have

5   thicker basal water (Schoof, 2010).

The results of changing the range of hydraulic conductivity ($K_m$), show little difference in the results at higher water volumes for the different runs. However, at 18 ka there is a big difference in the results. The results show that as hydraulic conductivity increases, the total water volume decreases. This is expected since increasing the conductivity increases the water flow and tunnel formation, allowing the water to evacuate from the ice sheet. The variation of $k_a$ and $k_b$ has little impact on

10   model results. This is rather fortuitous since they are not physical parameters that can be easily measured, whereas the range of hydraulic conductivity values can be constrained based on the type of sediment from field studies.

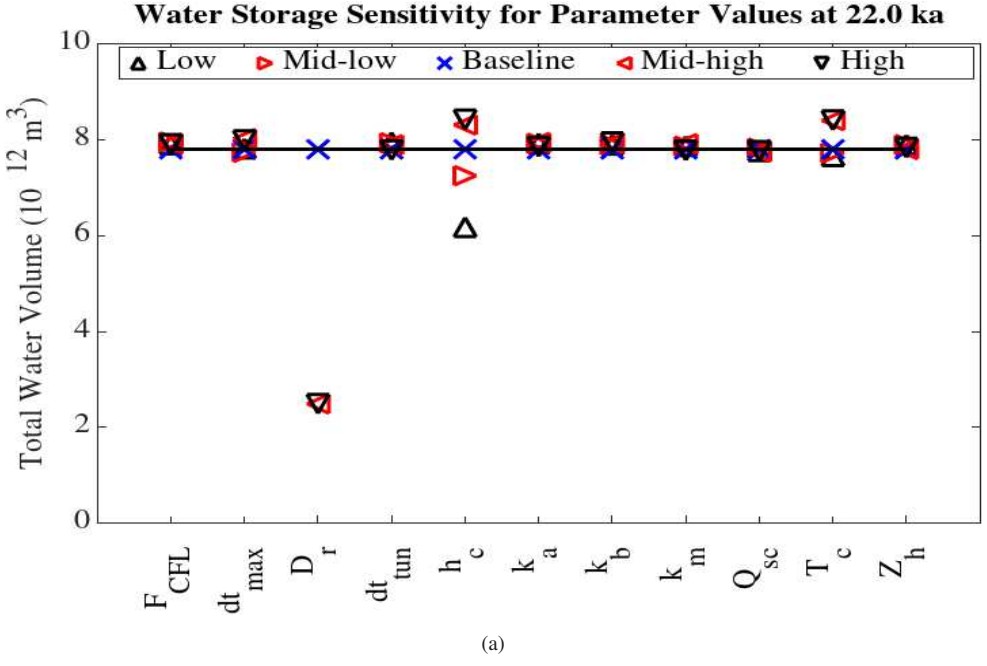

(a)

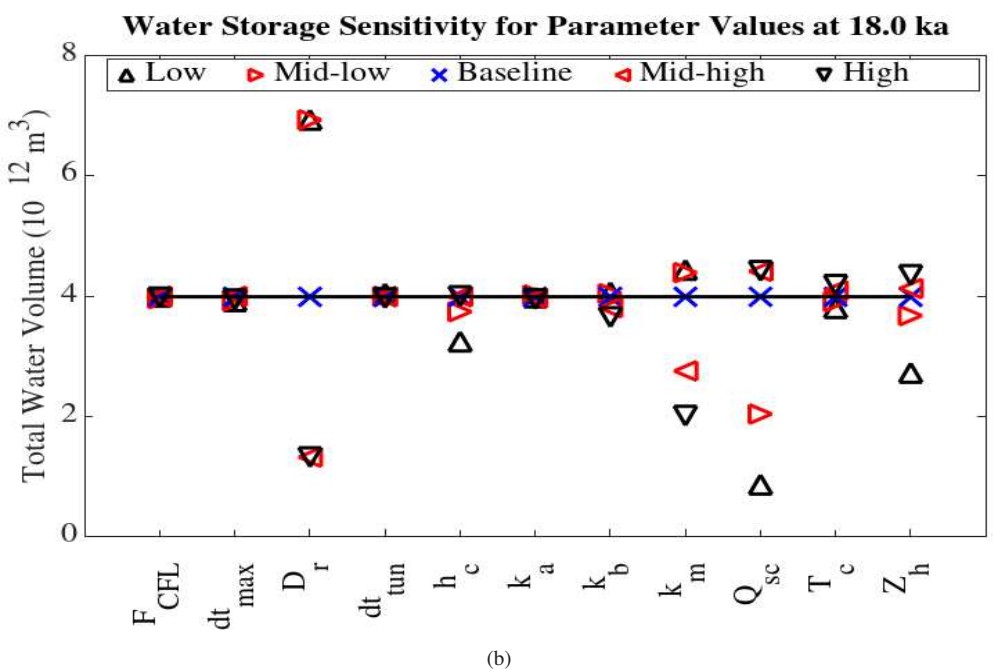

(b)

**Figure 6.** Sensitivity plot at a) 22 ka and b) 18 ka. Water storage for lowest $D_r$ value is off the scale ($221 \times 10^{12} m^3$ and $60 \times 10^{12} m^3$ for 22 and 18 ka respectively).

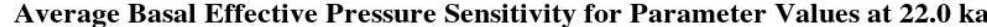

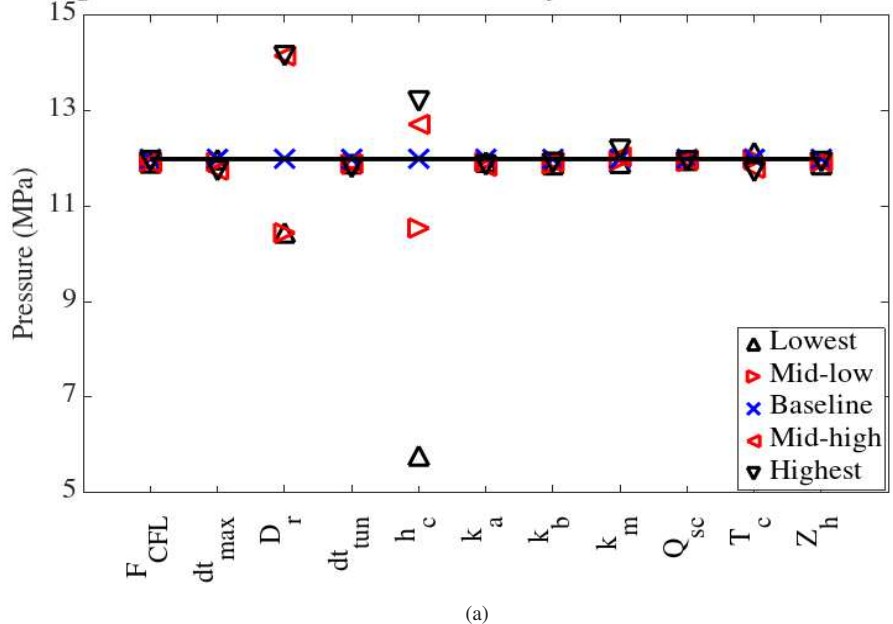

(a)

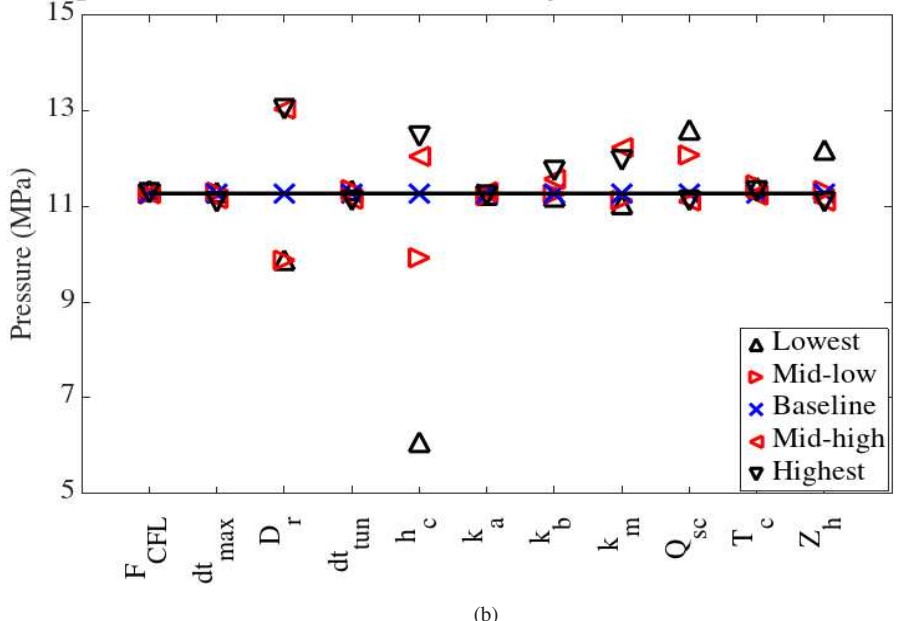

(b)

**Figure 7.** Sensitivity plot at a) 22 ka and b) 18 ka.

The plot of the average basal effective pressure, in Figure 7, shares similar properties to the water storage sensitivity in Figure 6. During periods of high water volume, the two most important parameters are the aquifer drainage and saturated sediment thickness. Their effects are much closer in terms of effective pressure due to the limiting of the pressure to ice overburden, thus limiting the effects of the aquifer drainage.

During the low water storage times, the other parameters become important to the basal effective pressure. The impact of saturated sediment thickness on basal effective pressure appears to be relatively insensitive to the amount of water storage, as the two plots in Figure 7 show similar results for both cases. Otherwise, the parameter values that lead to higher basal effective pressure are the same values that prevent the water from flowing out of the ice sheet in Figure 6.

Figures 6 and 7 both show the lack of importance of frequency of tunnel formation checks ($dt_{tun}$) which stems from the time scale of grid cell water refill (typically greater than the largest value for $dt_{tun}$). The effect of lowering $dt_{tun}$ may have a minor effect on when the tunnels form, but not how often.

One important test result is the low sensitivity of the average basal water thickness and effective pressure to the maximum allowable time step ($dt_{max}$). There is only a 10% water volume drop in the range of values chosen. Also, as the time steps become smaller (the lowest value was 1/3 of the baseline value), they begin to converge to an answer somewhere in the vicinity of the baseline values. This shows model stability and convergence for decreasing time steps.

As a caveat, these initial sensitivity tests likely hide spatially localized parametric sensitivities. More critically, feedbacks in a two-way coupled ice sheet and basal hydrology model configuration may strongly change relative sensitivities to basal hydrology parameters. These analyses will be better placed in a future study examining fully coupled dynamics.

## 7 Conclusions

This paper presents a physically-based hydrology model for numerical simulations over a glacial cycle at continental scales. The model considers two types of drainage systems: a distributed system that slowly drains basal water, and a fast draining channelized system. The distributed hydrology system is modelled with Darcy's law (Flowers, 2000) while the channelized system is likened to R-channels and solved using a fast down-gradient routing and lake solver (Tarasov and Peltier, 2006).

The model was tested over a set of synthetic ice profiles and topography. The results of these tests show that the model is mass conserving and that the water flows down the hydraulic potential gradient where it can exit the ice sheet or form subglacial lakes.

With the model validated using the synthetic ice sheets, the model was then one-way coupled to the GSM for testing on the North American Ice Complex at LGM. The sensitivity results in Figures 6 and 7 show that the significance of each parameter varies in time as the amount of basal water in the system changes. In times of high water input, the only significantly influential parameters are sediment pore space and aquifer drainage parameters. During times of lower water levels, other parameters begin to impact the basal water thickness and pressure as well. These parameters are related to tunnel formation, such as the bedrock bump height, tunnel criterion scaling factor, and the hydraulic conductivity.

The hydrology model also identified areas of low effective pressure, indicating areas of potentially fast flowing ice. These results were self-consistent with the GSM's parameterized areas of the fast-flowing ice.

The hydrology model presented here has been shown to be stable and robust for the range of parameters used in this study. The coupled model generally takes 5-8 hours to run for a North American glacial cycle ($0.5^o$ longitude by $1.0^o$ latitude resolution). This time includes the full GSM, suggesting that the hydrology model only contributes an hour or two of extra run-time over a full glacial cycle. The longest runs are those with the smallest time steps ($1/120$ year) or frequent calls to the tunnel solver, both of which show insignificant changes to the model results. This shows that the combined Heun's method and Leapfrog-trapezoidal scheme can be a viable numerical method for subglacial hydrology modelling.

As an initial implementation of a 2-D basal hydrology solver, there were several simplifications made to facilitate the initial study of the basic properties of the subglacial water dynamics. One simplification was the aquifer drainage parameter was used instead of a real aquifer drainage system (Flowers, 2000; Lemieux et al., 2008) which would provide a more realistic drainage and allow water to flow back into the subglacial system. The sediment thickness was simplified as a constant over the entire bed. Realistically, the sediment thickness would vary over different parts of the bed (e.g.; thinly-covered Canadian Shield bedrock as opposed to the thick cover of the prairies), as well as varying in time as the sediment cover changes due to sediment deformation (Melanson, 2012).

*Code availability.* Basal hydrology code with validation drivers is freely available on http://doi.org/10.5281/zenodo.1230046

## Appendix A: Model Numerics

### A1   Discretization of the Mass Balance Equation

The model uses the mass continuity equation (equation 2) for subglacial water. Expanding the divergence of the flux terms from the mass balance equation gives

$$\frac{\partial w}{\partial t} = \frac{1}{r\cos\theta}\left[\frac{\partial(Q_\phi)}{\partial\phi} + \frac{\partial(Q_\theta\cos\theta)}{\partial\theta}\right] + \dot{b} + d_{s:a} \tag{A1}$$

with $\theta$ representing the latitudinal direction, and $\phi$ representing the longitudinal direction.

Equation A1 is integrated over a finite-control volume

$$\iint \frac{\partial w}{\partial t}dV = \iint \left\{\frac{1}{r\cos\theta}\left[\frac{\partial(Q_\phi)}{\partial\phi} + \frac{\partial(Q_\theta\cos\theta)}{\partial\theta}\right] + \dot{b} + d_{s:a}\right\}dV \tag{A2}$$

using $dV = r^2\cos\theta d\phi d\theta$, equation A2 becomes

$$\iint \frac{\partial w}{\partial t}dV = \int_n^s \left\{\int_e^w \frac{\partial(Q_\phi)}{\partial\phi}d\phi\right\}rd\theta + \int_e^w \left\{\int_n^s \frac{\partial(Q_\theta\cos\theta)}{\partial\theta}d\theta\right\}rd\phi + \iint \left\{\dot{b} + d_{s:a}\right\}dV \tag{A3}$$

This then simplifies to

$$\iint \frac{\partial w_P}{\partial t} dV_P = \int_n^s \{Q_w - Q_e\} r d\theta + \int_e^w \{Q_s \cos\theta_s - Q_n \cos\theta_n\} r d\phi + \iint \left\{\dot{b} + d_{s:a}\right\} dV_P \tag{A4}$$

where the subscripts $n,e,s,w$ stand for north, east, south, and west interfaces respectively, and $P$ represents the central grid point.

Using the approximation $V_P = r^2 \cos\theta_P \Delta\phi\Delta\theta$, equation A4 can be approximated as

$$\begin{aligned}
\frac{\partial w_P}{\partial t} &= \frac{1}{r \cos\theta_P \Delta\theta}\{Q_s \cos\theta_s - Q_n \cos\theta_n\} \\
&+ \frac{1}{r \cos\theta_P \Delta\phi}\{Q_w - Q_e\} + \dot{b} + d_{s:a}
\end{aligned} \tag{A5}$$

## A2   Model Time-stepping

### A2.1   Predictor Time Steps

The model presented in this paper uses two predictor-corrector methods with different predictors but identical corrector. The first predictor method, based on Heun's method (Mathews and Fink, 2004), is used when only the current water thickness field is self-consistently available. It generates the first time step during each call of the basal hydrology subroutine, and for any grid cell that has been activated as a tunnel just prior to the current Darcy flow time step. The first of these two conditions could be easily amended to just the first timestep of the whole model run if desired (and would be required for the case of synchronous

model coupling). We chose the current formulation to simplify module coupling with configurations that involved multiple ice sheets on separate grids.

The first step in Heun's method is to take some initial conditions ($w_P^m$), and do an Euler Forward time step,

$$w_P^{(m+1)*} = w_P^m + \frac{\Delta t}{r \cos\theta_P}\left\{\frac{Q_s^m \cos\theta_s - Q_n^m \cos\theta_n}{\Delta\theta} + \frac{Q_w^m - Q_e^m}{\Delta\phi}\right\} + (\dot{b}^0 + d_{s:a}^0)\Delta t \tag{A6}$$

where $w_P^{(m+1)*}$ is the tentative (predicted) value for the next time step. The source terms ($\dot{b}^0 + d_{s:a}^0$) do not change within a

call to BrAHMs and therefore retain the $0$ time index (a positivity constraint ensures that $w_P \geq 0$).

When the previous timestep value of $w_P$ is self-consistently available, we use the second order accurate Leapfrog-Trapezoidal scheme (the Heun scheme is only first order accurate). The Leapfrog predictor for this scheme calculates the intermediate predicted value for the next time step, $w^{(m+1)*}$, as

$$w_P^{(m+1)*} = w_P^{m-1} + \frac{2\Delta t}{r \cos\theta_P}\left\{\frac{Q_s^m \cos\theta_s - Q_n^m \cos\theta_n}{\Delta\theta} + \frac{Q_w^m - Q_e^m}{\Delta\phi}\right\} + 2(\dot{b}^0 + d_{s:a}^0)\Delta t \tag{A7}$$

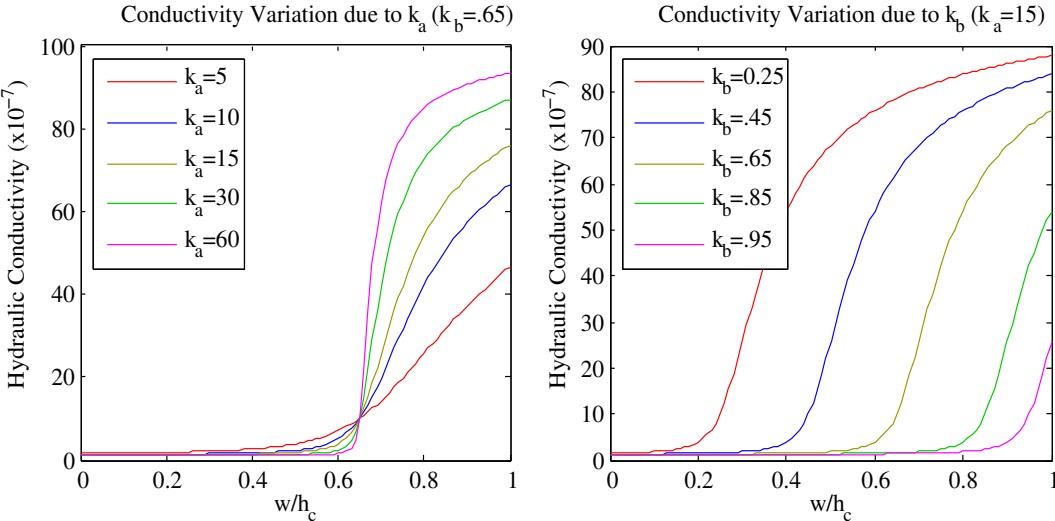

**Figure A1.** Variations of subglacial hydraulic conductivity $K$ with respect to changes in hydraulic parameters $k_a$ and $k_b$ for values of $K$ ranging between $1.0 \times 10^{-7}$–$1.0 \times 10^{-5}$.

### A2.2 Trapezoidal Corrector

Regardless of which predictor equation is active, the trapezoidal scheme is applied to give the corrected value, $w^{m+1}$, as

$$w_P^{m+1} = w_P^m + \frac{\Delta t}{2r\cos\theta_P} \left\{ \frac{Q_w^{(m+1)^*} - Q_e^{(m+1)^*}}{\Delta\phi} + \frac{Q_s^{(m+1)^*}\cos\theta_s - Q_n^{(m+1)^*}\cos\theta_n}{\Delta\theta} \right. \tag{A8}$$
$$\left. + \frac{Q_w^m - Q_e^m}{\Delta\phi} + \frac{Q_s^m\cos\theta_s - Q_n^m\cos\theta_n}{\Delta\theta} \right\} + (\dot{b}_s^0 + d_{s:a}^0)\Delta t$$

### A3 Discretization of the Darcian Flux

The Darcian flux, $Q$, is given in equation 3. The values for hydraulic conductivity, basal water thickness, and pressure, along with bed topography are calculated at the grid cell centres. To calculate $Q$ at the grid cell interfaces, the aforementioned values must be assigned values at the interfaces.

If we consider the case of the flux on the westward edge of the grid cell, $Q_w$, then the pressure gradient ($\nabla\{P + \rho_w g z_b\}$ from equation 3) is simply the difference between the pressure values at the grid cell centres

$$Q_w = \frac{Kw}{\rho_w g} \frac{P_W - P_P + \rho_w g(z_{b_W} - z_{b_P})}{r\cos(\theta_P)\Delta\phi} \tag{A9}$$

where the $W$ subscript indicates the value of the grid point to the west of the central point, and the $P$ subscript represents the grid cell of interest.

Following the rules of Patankar (1980), the hydraulic conductivity at the grid cell interface is set to the geometric mean of the values at the adjacent grid cell centres

$$Q_w = \left( \frac{2 K_W K_P}{K_W + K_P} \right) \frac{w}{\rho_w g} \frac{P_W - P_P + \rho_w g (z_{b_W} - z_{b_P})}{r \cos(\theta_P) \Delta \phi} \tag{A10}$$

To simplify the flux equation, the upwind scheme (Patankar, 1980) was used to give the value of the water at the interface
(i.e.; the value of $w$ is equal to the water thickness of the grid cell with the highest pressure). This gives the final equation of the flux as

$$Q_w = \left( \frac{2 K_W K_P}{K_W + K_P} \right) \left( \frac{1}{\rho_w g r \cos(\theta_P) \Delta \phi} \right) \left[ \max\{ w_W [P_W - P_P + \rho_w g (z_{b_W} - z_{b_P})], 0 \} \right. \tag{A11}$$
$$\left. - \max\{ -w_P [P_W - P_P + \rho_w g (z_{b_W} - z_{b_P})], 0 \} \right]$$

where $Q_w$ is positively defined if water flows eastward into the centre grid cell. Likewise all the other fluxes can be defined in
a similar fashion. Outgoing fluxes are limited to ensure positive basal water thickness.

## A4   Flowchart of Model Procedure

*Competing interests.* There are no competing interests.

*Acknowledgements.* We thank J. Johnson and J. Seguinot for thoughtful and helpful reviews. This work also significantly benefited from a review of the associated Thesis by J. Johnson. Entcho Demirov offered some helpful numerical suggestions. This work was supported by a
NSERC Discovery Grant (LT), the Canadian Foundation for Innovation (LT), and the Atlantic Computational Excellence Network (ACEnet).

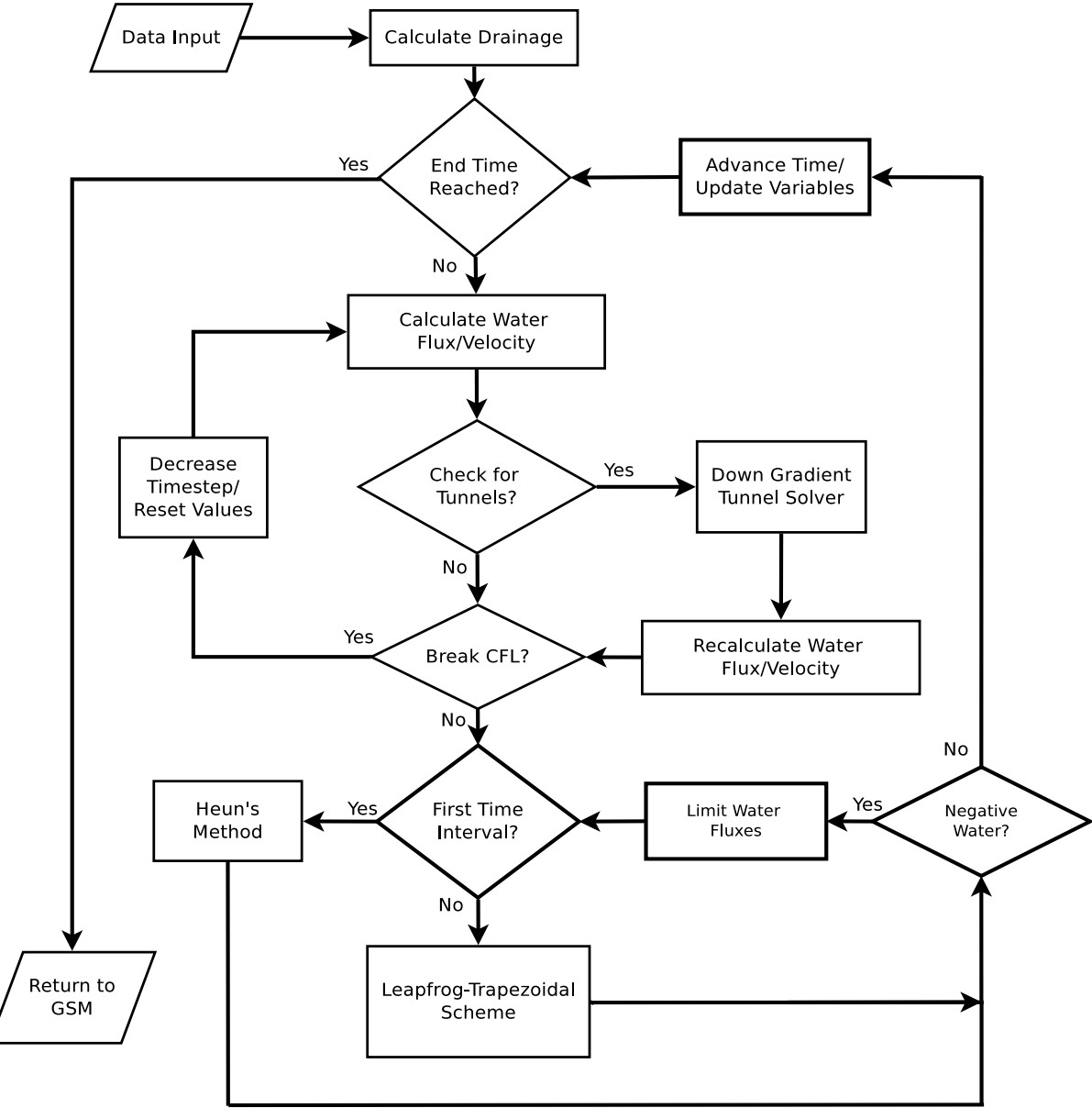

**Figure A2.** Hydrology model flow chart highlighting the processes involved in simulating basal water flow.

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
