# Peer review of "BrAHMs V1.0: A fast, physically-based subglacial hydrology model for continental-scale application."

_Geoscientific Model Development, 2017_

## Short Comment (SC1) · 13 Dec 2017

As explained in https://www.geoscientific-model-development.net/about/manuscript_types.html GMD is encouraging that authors upload the program code of models (including relevant data sets) as a supplement or make the code and data or the exact model version described in the paper accessible through a DOI (digital object identifier). In case your institution does not provide the possibility to make electronic data accessible through a DOI you may consider other providers (eg. zenodo.org of CERN) to create a DOI. Please note that in the code accessibility section you can still point the reader how to obtain the newest version.

[Figure]

If for some reason the code and/or data cannot be made available in this form (e.g. only via e-mail contact) the"Code Accessibility" section need to clearly state the reasons for why access is restricted (e.g. licensing reasons).

Lutz Gross GMD Executive Editor

---

## Referee Comment (RC1) · J.V. Johnson (Referee) · 30 Dec 2017

This paper presents a model for subglacial hydrology suitable for continental scale ice sheets. Here, concerns are different from the glacier scale hydrologic models that have seen significant change in recent years. Time steps are longer (days vs hours) and spatial scale is larger (tens of kilometers vs hundreds of meters). The paper is novel in that it defines a set of physical processes related to channelized and distributed flow that can be efficiently and robustly solved using numerical methods. After describing the model, the authors conduct experiments on an idealized, parabolic ice sheet on a flat bed, and conclude with a sensitivity study conducted on a reconstruction of the

[Figure]

North American ice sheet complex during the last glacial maximum (18 and 22 kybp).

The model represents worthwhile contribution to the literature because its lower fidelity physics are well suited to the problem ice sheet reconstruction via simulation of 100 ky glacial cycles. Before it is ready for publications, I see a number of issues for the authors to address. I believe that they are significant enough that I've called them 'major' - mostly to assure that something is done to address them. In short, my primary criticism is that I do not think that the results are reproducible because important aspects of model setup are omitted. I also think the work should be better scoped so that readers understand the distinctions between this model and other, recent works related to subglacial hydrology.

* The simulations, especially those having to do with the LGM (last glacial maximum), have to be described in more detail. Enough is missing that I'm struggling to evaluate the conclusions of the paper. In particular: How are model runs set up, and how does the ISM (ice sheet model) get to the point where hydrology is called? I hoped that citing some of the Tarasov's prior work could be used, but didn't find it. I understand that this is a 'one-way' coupling (ISM forces basal hydro), but that is not sufficient. What are the fields that force the basal hydro model (ice sheet thickness, basal temperature, and basal melt rate?) How is the melt-rate computed? Is melt on the surface of the ice sheet routed to the bed? How is basal traction determined in the absence of two-way coupling? Finally, see my next point on the stability of a nonlinear system. This is perhaps my greatest concern.

* The system of equations includes a number of strong non-linearities in terms of the key prognostic variable - w (effective water depth). Specifically we have

** Flux, Q, depends on w and K, conductivity, which has w dependence ** Water pressure, or the potential surface that water is routed down, depends on w ** A critical flux, dependent upon w, can have a rapid and strong impact on w

What is the interrelation between the time stepping of the ISM and the basal hydrology

model? Does the hydrology model achieve steady state between ISM updates? If not, are the larger changes in the potential field on ISM time steps sufficient to produce shocks to the transient hydromodel? Are these shocks 'captured' in a numerically robust way? How does the rapid drainage mechanism, and its impact on the effective pressure, impact the system? Does it give rise to rapid oscillations in streaming behavior? Are any of these non-linear couplings and effects sensitive to the spatial/temporal discretization? If the ISM is forcing the hydromodel at each ISM time step - then how sensitive is the hydro model to the initial conditions? In particular the distribution of basal water. There are mentions of stability and robust solutions in the text, but they are just that - mentions. I'd like to see more on this, to assure the reader the results are stable across discretizations of space and time.

* Continuing with the issue of reproducibility, the code should be more accessible. Publication should include a URL repository where the code can be accessed. Tag the branch used in the publication.

* The distinctive features of this model need to be contrasted to the wealth of recent publications in the area of subglacial hydrology. (eg Schoof, Werder, Hewitt, and Hoffmann). There is a need for a continental scale model like this, but it should be established by documenting how and why other models are not suitable to this task. Similarly, the authors claim that other continental scale models do not include sub-glacial hydrology. I don't think this is true. PISM has some accounting for basal water, and so does SICPOLIS. Pollard and DeConto treat hydrology as it relates to sediment. ISSM and Elmer ICE both have hydrology models. CISM contains an ISM, and it contains some subglacial hydrological components developed by Hoffman. It is possible that none of these are good tools for continental/glacial cycle scale studies that are the specialty of Tarasov, but this should be argued persuasively in the paper. Much more should be done here.

* Axially symmetric experiments should be presented with bivariate plots. Not much is gained by inspecting these highly symmetric solutions. Something is lost in the color

map, which might hide high frequency oscillations in the solution.

* The focus for the sensitivity study should be streaming behavior, that is the point of the hydro model. Averaging quantities across the entire ice sheet diminishes the importance of changes to parameters. Why not consider the impact of parameter changes to a set of grid cells that are characterized by low effective pressure at 18 and 22 kybp?

In general, the paper is well written and logically presented. I had some minor notes about the choice of words, but given the need for major revisions, it's probably best to wait for those revisions before picking apart the language. Concluding, I'd like to see this paper published. The work occupies a unique niche in a world where basal hydrology models continue to add complexity in the absence of observation. It's refreshing to return to fundamentals and basic physical processes. I believe my concerns are specific, and can be addressed in a reasonable time frame. I look forward to seeing a revised manuscript.

---

## Referee Comment (RC2) · J. Seguinot (Referee) · 13 Feb 2018

I would like to apologize to the authors for this much delayed review.

M. Kavanagh and L. Tarasov present a new model to compute water flow under ice sheets, and study feedback processes between subglacial water flow and the much slower dynamics of overlying glacier ice. Thus, the model physics and numerics have been tailored for coupling to ice sheet models which typically operate on continental (thousands of kilometres), and glacial cycle (multi-millenial) scales that characterise the spatio-temporal evolution of the Earth's largest flowing ice masses.

[Figure]

The need for coupled models of ice dynamics and subglacial hydrology has been identified for decades, however it has been subject to two major limitations. First, subglacial water flows much faster than glacier ice, which is an issue for both physical and numerical model implementations. Second, although subglacial hydrology theories are available, physical parameters are largely unconstrained due to the difficulty of observations. In the present manuscript, M. Kavanagh and L. Tarasov address these issues by using simplified physics, a semi-implicit discretization scheme, and a parameter sensitivity study.

The paper contains an introduction summarizing recent advances in modelling subglacial hydrology, a description of the model's physics, an application to a synthetic test case where the model yields expected results, and a more realistic application on the modelled Last Glacial Maximum and early deglacial North American ice sheet complex, including a sensitivity study to the most important model parameters. Discretization schemes for subglacial hydrology are explicated in Appendix.

BrAHMs is coupling subglacial hydrology model to ice dynamics in ways that will facilitate its application to continental-scale ice sheet dynamics. Publication of the model physics, numerics, and the presented test cases in *Geoscientific Model Development* makes a lot of sense and I fully support it. Nevertheless, I am concerned by the fact that source code has not yet been made publicly available, and I think that the manuscript need a few crucial changes before publication in order to ensure reproducibility.

Below I provide comments regarding the points for which I believe changes will improve the manuscript. I hope the authors will find these helpful in revising their manuscript and wish them success with final publication and future applications of this innovative model.

[Figure]

**1 General comments**

**Code availability**

I think it is policy of *Geoscientific Model Development* that all computing code accompanying publications should be made publicly available, unless reasons against that are clearly stated. Since BrAHMs is one of the first subglacial hydrology models allowing coupling to an ice sheet model, I think code publication would be strongly beneficial to both the authors and the ice sheet modelling community.

Actually I would even recommend a platform that allows version control and issue tracking. For instance PISM (https://github.com/pism/pism) is another coupled ice dynamics and subglacial hydrology model for which source code publication and public bug tracking has been highly beneficial.

**Parameter values**

In the present manuscript, Table 1 lists hydrological parameter ranges used in the sensitivity test. However, values for parameters kept fixed in the sensitivity test are not given in the manuscript. These include glacial system model parameters (Eq. 1), subglacial hydrology model fixed parameters (Eqs. 2–5), and parameters defining the synthetic ice surface geometries and melt distributions for the first test case (Eqs. 6–8). For instance, the scale of the synthetic ice sheet and the amplitude of bed perturbations used in the test case are crucial information currently missing from the manuscript.

For the sake of reproducibility, including future reproduction of the synthetic test case by other models, I think all parameter values should be included in the manuscript before publication. I would suggest a separate table containing all fixed parameter values.

**Readability of figures**
I found that the current figures don't reflect the scientific quality of the work undertaken by the authors. This is especially destructive given that the manuscript text is actually very well written. Below I suggest a few simple changes that I believe will enhance the readability of figures.

On Figs. 1–3, the choice of colours does not serve the results at all. Since non-null water thickness and pressure is localized around the ice sheet margins, it is very hard to discern the individual colour bands. Instead I would suggest monochromatic (e.g. white-to-blue, white-to-red) colourmaps, preferably different for water thickness and pressure.

Also on Figs. 1–3, contour lines are so thin that they became invisible on my print. Overlayed basal and surface topography contours (Fig. 1b) are also hard to distinguish. I suggest to remove basal topography contours, and slightly thicken surface topography contours.

Finally, Figs. 4–5 are hard to read because many markers overlap. Here my suggestion would be a different presentation, using volume errors instead of total water volume, a logarithmic scale to discern small errors, and perhaps different colours for positive and negative errors.

**2   Specific Comments**

**p. 2, l. 4–7**: Only a few subglacial hydrology models have been described in the literature for continental-scale ice sheets. [...] These models take various approaches to simulate the flow of basal water using physically-based equations.

This paragraph introduces a short review of recent developments in subglacial hydrology modelling (decoupled from ice dynamics), which I found very useful to guide the

reader in understanding choices made by the authors in designing their own model. However, one of the first questions I had when opening the manuscript was how BrAHMs differs from the approach employed by Bueler and van Pelt (2015), to my knowledge the first functional model of coupled subglacial hydrology and ice sheet dynamics. I think this review is be the right place to address this point.

**p. 4, l. 27–28**: We use an empirical relation for water pressure from Flowers (2000).

Here I think it would help to shortly explain the type of measurements and time scale used to develop this empirical relation (Eq. 4), or at least give a page number.

**p. 5, l. 1–2**: $P$ is limited to ice overburden pressure. $h_c$ equals till thickness times porosity and is effectively the water thickness that the till can hold before becoming over-saturated.

I understand that $P$ is capped at overburden, but additional water could be stored in the till, resulting in $w > h_c$. Is this correct? A short sentence to clarify what is happening over saturation would help here.

**p. 5, l. 11–14**: From here the model employs a down hydraulic gradient solver (Tarasov and Peltier, 2006) that looks at the neighbours of a tunnel cell and allows water to flow instantaneously down the path of steepest potential gradient (channelizing cells along that path) until there is no cell with a lower hydraulic potential (forms subglacial lake) or the water exits the ice sheet.

I assume that the down gradient solver is the computational bottleneck of the model. I guess that 'instantaneously' means that the hydrological solver is ran offline while the ice model pauses. Here it would help to clarify whether that is the case (or if only the tunnel solver is ran offline at regular intervals), and (already here in the methods) how often would one presumably need to run the hydrology model, and what are the

consequences of this assumption in terms of the domain of application of the coupled model.

**p. 6, Figs. 1 caption**: The symmetric results in a) are due to a known issue with the tunnel solver being slightly asymmetric.

The results in a) look symmetric indeed, but I wonder if the authors meant to write about the asymmetric results in b), which would make more sense.

**p. 6, Figs. 1–2**:

Eq. 6–8 are given in polar (or at least radius) coordinate. The discretization perfomed in the appendix also uses polar coordinates. However Figs. 1–2 appear to use a regular grid. Labelling the x and y axes would help to resolve this ambiguity.

**p. 7, l. 7**: The next test placed an ice sheet flattened near the edges on a dilating (sinusoidally-wavy) bed.

Could the authors please include a formula for the sinusoidally-wavy bed?

**p. 7, l. 20**: Next, the ice dome was placed on an incline to test the flow of water.

Could the authors please include a formula for the inclined bed?

**p. 8, l. 5–6**: The simplified aquifer drainage of Johnson (2002), uses an aquifer that simply drains a percentage of the present water in a cell. The percentage of water drained in this model is represented by the $D_r$ parameter.

I wonder how this new parameter $D_r$ relates to $d_{s:a}$ (Eq. 2).

**p. 8, l. 12–13**: The water flux between cells is directly proportional to the hydraulic conductivity of the sediment. For each run, the conductivity was allowed to vary between
a minimum and maximum value defined in the range of $K_m$.

Here it would be nice to have a short explanation as to why this approach (Eq. 9) is superior to a constant conductivity, and whether it is backed up by measurements or theory.

**p. 9, l. 4–8**: In the hydrology model, this is represented by parameter $T_c$ , which acts to reduce the conductivity as a function of temperature. When the basal temperature is close to the pressure melting point (PMP), there is little change in the hydraulic conductivity. Conductivity decreases to an extremal low value as the temperature approaches the value of $T_c$.

I wonder if the decrease in conductivity could be described by a function? An equation would be very useful here.

**p. 14, l. 1–2**: The results of these tests show that the model is mass conserving.

This is not very obvious from the rest of the manuscript. I would suggest to add a plot of mass conservation errors (claimed on the order of $10^{-12}$ m) or remove this statement.

**3 Technical Comments**

**p. 1, Affiliations**: St. Johns / St John's

The spelling probably needs to be homogenised here.

**p. 1, l. 13–14**: Channel formation [...] display the arborescent

An 's' is missing here.

**p. 7, l. 30–31**: As such, there are a number of poorly constrained parameters in the model.

This is a good place to reference to Table 1.

**References**

Bueler, E. and van Pelt, W.: Mass-conserving subglacial hydrology in the Parallel Ice Sheet Model version 0.6, Geosci. Model Dev., 8, 1613–1635, doi:10.5194/gmd-8-1613-2015, 2015.

---

## Author Comment (AC1) · 26 Apr 2018

We thank both reviewers for their thoughtful evaluation. Our detailed response and latexdiff of the revised manuscript is in the attached pdf.

Please also note the supplement to this comment: https://www.geosci-model-dev-discuss.net/gmd-2017-275/gmd-2017-275-AC1-supplement.pdf

---

## Author Response (AR1)

**Interactive comment on BrAHMs V1.0: A fast, physically-based subglacial hydrology model for continental-scale application**

Mark Kavanagh and Lev Tarasov

April 26, 2018

**1 Appreciation and One Brief Note on Changes to Paper**

We thank both reviewers for their thoughtful and detailed reviews.

Before addressing the issues raised, it should be noted that the validation tests have been changed from a parabolic bed to a normally-distributed profile since the initial submission. The flatter edges of the normally-distributed bed allow for better analysis of lake formation (as discussed in the updated paper).

**2 Discussion with Dr. J. Seguinot**

**Referee** I would like to apologize to the authors for this much delayed review. M. Kavanagh and L. Tarasov present a new model to compute water flow under ice sheets, and study feedback processes between subglacial water flow and the much slower dynamics of overlying glacier ice. Thus, the model physics and numerics have been tailored for coupling to ice sheet models which typically operate on continental (thousands of kilometres), and glacial cycle (multi-millenial) scales that characterise the spatio-temporal evolution of the Earths largest flowing ice masses.

The need for coupled models of ice dynamics and subglacial hydrology has been identified for decades, however it has been subject to two major limitations. First, subglacial water flows much faster than glacier ice, which is an issue for both physical and numerical model implementations. Second, although subglacial hydrology theories are available, physical parameters are largely unconstrained due to the difficulty of observations. In the present manuscript, M. Kavanagh and L. Tarasov address these issues by using simplified physics, a semi-implicit discretization scheme, and a parameter sensitivity study.

The paper contains an introduction summarizing recent advances in modelling sub- glacial hydrology, a description of the models physics, an application to a synthetic test case where the model yields expected results, and a more realistic application on the modelled Last Glacial Maximum and early deglacial North American ice sheet complex, including a sensitivity study to the most important model parameters. Discretization schemes for subglacial hydrology are explicated in Appendix.

BrAHMs is coupling subglacial hydrology model to ice dynamics in ways that will facilitate its application to continental-scale ice sheet dynamics. Publication of the model physics, numerics, and the presented test cases in Geoscientific Model Development makes a lot of sense and I fully support it. Nevertheless, I am concerned by the fact that source code has not yet been made publicly available, and I think that the manuscript need a few crucial changes before publication in order to ensure reproducibility.

Below I provide comments regarding the points for which I believe changes will improve the manuscript. I hope the authors will find these helpful in revising their manuscript and wish them success with final publication and future applications of this innovative model.

I think it is policy of Geoscientific Model Development that all computing code accompanying publications should be made publicly available, unless reasons against that are clearly stated. Since BrAHMs is one of the first subglacial hydrology models allowing coupling to an ice sheet model, I think code publication would be strongly beneficial to both the authors and the ice sheet modelling community.

Actually I would even recommend a platform that allows version control and issue tracking. For instance PISM (https://github.com/pism/pism) is another coupled ice dynamics and subglacial hydrology model for which source code publication and public bug tracking has been highly beneficial.

**Authors** The senior author (Lev Tarasov) has to date relied on offering open code availability via email queries which has the added benefit of making it easier to help potential users. GDM policy also permits this (though prefers availability via a public server). And our submitted paper clearly stated that the BrAHMs source code is freely available upon request. However, given that the referee found this inadequate, we have now deposited the BrAHMs code on a public server (http://doi.org/10.5281/zenodo.1230046).

**Referee** n the present manuscript, Table 1 lists hydrological parameter ranges used in the sensitivity test. However, values for parameters kept fixed in the sensitivity test are not given in the manuscript. These include glacial system model parameters (Eq. 1), subglacial hydrology model fixed parameters (Eqs. 25), and parameters defining the synthetic ice surface geometries and melt distributions for the first test case (Eqs. 68). For instance, the scale of the synthetic ice sheet and the amplitude of bed perturbations used in the test case are crucial information currently missing from the manuscript.

For the sake of reproducibility, including future reproduction of the synthetic test case by other models, I think all parameter values should be included in the manuscript before publication. I would suggest a separate table containing all fixed parameter values.

**Authors** As requested, tables have been added that list the BrAHMs parameters used in this paper's analysis. Table 1 contains a list of the parameters used to describe the synthetic ice sheet from equations 5 - 8. Table 2 lists the baseline values for the hydrology parameters that are used in the validation and GSM modelling sections.

Note, we presume the reviewer was just refering to BrAHMs parameters. The main focus of this paper is on the development of a new subglacial hydrology. The cited references for the GSM describe the GSM in much more appropriate detail. The GSM has dozens of parameters and concepts that would weight down this paper and remove focus from the hydrology model.

To require inclusion of all GSM parameters would be akin to requiring every GCM based modelling paper to list the litterally 100's of parameters in a regular GCM.

However, as detailed in the response to Dr. Johnson's comments below, we have added a bit more information about how basal sliding is implemented.

**Referee** I found that the current figures dont reflect the scientific quality of the work undertaken by the authors. This is especially destructive given that the manuscript text is actually very well written. Below I suggest a few simple changes that I believe will enhance the readability of figures.

On Figs. 13, the choice of colours does not serve the results at all. Since non-null water thickness and pressure is localized around the ice sheet margins, it is very hard to discern the individual colour bands. Instead I would suggest monochromatic (e.g. white-to-blue, white-to-red) colourmaps, preferably different for water thickness and pressure.

Also on Figs. 13, contour lines are so thin that they became invisible on my print. Overlayed basal and surface topography contours (Fig. 1b) are also hard to distinguish. I suggest to remove basal topography contours, and slightly thicken surface topography contours.

Finally, Figs. 45 are hard to read because many markers overlap. Here my suggestion would be a different presentation, using volume errors instead of total water volume, a logarithmic scale to discern small errors, and perhaps different colours for positive and negative errors.

**Authors** The figures in the paper have been modified to be clearer, but not with colour scheme suggested, as we find it does not best present the data. The standard high contrast "Jet" colour scheme is now used where red is meant to show high water levels or high basal pressure (low effective pressure), which are likely to be areas of fast flowing ice (were the GSM and BrAHMs fully coupled).

The contours have been made thicker, and the font size was increased to improve readability. We retained the linear scale on the sensitivity plots (formerly 4-5, now 5-6) as this clearly conveys the relevant information. The overlapping markers indicate that the model is insensitive to a particular parameter, as discussed in the text. Extracting minute differences via a logarithmic grid would erroneously convey significance in differences where currently plot symbols overlap.

**Referee** p. 2, l. 47: "Only a few subglacial hydrology models have been described in the literature for continental-scale ice sheets. [...] These models take various approaches to simulate the flow of basal water using physically-based equations."

This paragraph introduces a short review of recent developments in subglacial hydrology modelling (decoupled from ice dynamics), which I found very useful to guide the reader in understanding choices made by the authors in designing their own model. However, one of the first questions I had when opening the manuscript was how BrAHMs differs from the approach employed by Bueler and van Pelt (2015), to my knowledge the first functional model of coupled subglacial hydrology and ice sheet dynamics. I think this review is be the right place to address this point.

**Authors** Good point. That paper is now cited and a discussion has been added on how BrAHMs and the model of Bueler and van Pelt (2015) differ. For the context of glacial cycle integrations, the

key differences are the inclusion of channelized drainage in BrAHMs and a different treatment for determining pressure in Bueler and van Pelt.

**Referee** p. 4, l. 27-28: "We use an empirical relation for water pressure from Flowers (2000)."

Here I think it would help to shortly explain the type of measurements and time scale used to develop this empirical relation (Eq. 4), or at least give a page number.

**Authors** A page number has been added to the reference that can guide interested readers to the source more quickly, as the original author(Dr. Gwen Flowers) provides more detail in developing the pressure equation.

Also, the following text has been added

> "Flowers (2000) derived this equation by considering sub-grid variation in bed elevation and associated sediment thickness (and therefore water thickness, all for the context of 40 m x 40 m grid-cell modelling of Trapridge Glacier). A further consideration (without the overburden limit) was that the nonlinearity would address dynamic adjustments in porosity and prevent stiffness in the dynamic equations caused by unrealistic heterogeneity in the modelled water distribution. Though derived for glacier-scale flow through a heterogenous macroporous sediment layer, our working hypothesis is that this empirical relation approximately captures large scale pressure response for subglacial distributed flow through any heterogenous structure (be it a mix of cavities of different size, patchy sediment, Nye channels...). The limiting of water pressure to overburden is justified by the low likelihood of water pressure exceeding the overburden pressure for any significant amount of time on glacial cycle timestep scales."

**Referee** p. 5, l. 12: "$P$ is limited to ice overburden pressure. $h_c$ equals till thickness times porosity and is effectively the water thickness that the till can hold before becoming over-saturated."

I understand that P is capped at overburden, but additional water could be stored in the till, resulting in $w > h_c$. Is this correct? A short sentence to clarify what is happening over saturation would help here.

**Authors** Correct. Edits to the text were made to clarify this:

> "where $P_I$ is the ice overburden pressure. $P$ is limited to ice overburden pressure when $w \geq h_c$. Saturated sediment water thickness, $h_c$, equals till thickness times porosity and is effectively the water thickness that the till can hold before becoming over-saturated (at which point the excess water is stored between the till and the ice)."

**Referee** p. 5, l. 1114: "From here the model employs a down hydraulic gradient solver (Tarasov and Peltier, 2006) that looks at the neighbours of a tunnel cell and allows water to flow instantaneously down the path of steepest potential gradient (channelizing cells along that path) until there is no cell with a lower hydraulic potential (forms subglacial lake) or the water exits the ice sheet."

I assume that the down gradient solver is the computational bottleneck of the model. I guess that 'instantaneously means that the hydrological solver is ran offline while the ice model

pauses. Here it would help to clarify whether that is the case (or if only the tunnel solver is ran offline at regular intervals), and (already here in the methods) how often would one presumably need to run the hydrology model, and what are the consequences of this assumption in terms of the domain of application of the coupled model.

**Authors** Dr. Seguinot comments indicate that there may be some confusion as to when and how the down gradient solver is implemented. This issue has been addressed by the revised text:

> "To simulate the change between different drainage systems, at regular user-defined intervals, the channel flow subroutine is called. Grid cells for which the water flux exceeds the distributed flow stability criterion (equation 5) are marked as tunnel grid cells. A down hydraulic gradient solver (Tarasov and Peltier, 2006) instantaneously[1] moves water down the path of steepest potential gradient (channelizing grid cells along that path) until there is no grid cell with a lower hydraulic potential (so forms subglacial lake) or the water exits the ice sheet. The solver considers all adjacent grid cells (including corner adjacency) when searching for the lowest hydraulic potential. Once the tunnel water transport is complete, the tunnels are assumed closed and the distributed flow algorithm continues."

The frequency at which the tunnel solver is called is subject to sensitivity analysis via variations in $dt_{tun}$.

**Referee** p. 6, Figs. 1 caption: "The symmetric results in a) are due to a known issue with the tunnel solver being slightly asymmetric."

The results in a) look symmetric indeed, but I wonder if the authors meant to write about the asymmetric results in b), which would make more sense.

**Authors** Indeed. Fixed as suggested by referee.

**Referee** p. 6, Figs. 12:
Eq. 6 – 8 are given in polar (or at least radius) coordinate. The discretization perfomed in the appendix also uses polar coordinates. However Figs. 12 appear to use a regular grid. Labelling the x and y axes would help to resolve this ambiguity.

**Authors** To clear up any confusion between polar and Cartesian grids, the accompanying text to the figures has been modified to include the following text:

> "It should be noted that the model is based on spherical polar coordinates (as it is designed for modelling large sections of the Earth's surface), and so the figures presented here are akin to the Mercator projection..."

**Referee** p. 7, l. 7: "The next test placed an ice sheet flattened near the edges on a dilating (sinusoidally-wavy) bed."

Could the authors please include a formula for the sinusoidally-wavy bed?

p. 7, l. 20: "Next, the ice dome was placed on an incline to test the flow of water."

Could the authors please include a formula for the inclined bed?
* * *
[1] During the tunnel flow, no model time is stepped as tunnel drainage is computed diagnostically.

**Authors** As suggested by Dr. Seguinot, equations for the bed have been added (equations 9 and 10)

**Referee** p. 8, l. 56: "The simplified aquifer drainage of Johnson (2002), uses an aquifer that simply drains a percentage of the present water in a cell. The percentage of water drained in this model is represented by the $D_r$ parameter."

I wonder how this new parameter $D_r$ relates to $d_{s:a}$ (Eq. 2)

**Authors** A note has been made in Section 6.1, where $D_r$ is introduced, that $d_{s:a} = D_r * w$.

**Referee** p. 8, l. 1213: "The water flux between cells is directly proportional to the hydraulic conductivity of the sediment. For each run, the conductivity was allowed to vary between a minimum and maximum value defined in the range of $K_m$."

Here it would be nice to have a short explanation as to why this approach (Eq. 9) is superior to a constant conductivity, and whether it is backed up by measurements or theory.

**Authors** New text has been added to explain the origin of the hydraulic conductivity equation.

> "which is a constitutive equation of the logarithmic form of the upstream area of a grid cell (Flowers, 2000, page 80). Data from Trapridge Glacier shows a similar relation between upstream area and water pressure as the hydraulic conductivity equation (a high and low regime with a transition zone). Flowers (2000) assumes that the upstream area is related to the connectivity in the grid cell (the more connected a grid cell is, the more upstream area it should have). This would suggest that the hydraulic conductivity (its connectivity) is dependent on the water level, and is of the form of equation 12."

**Referee** p. 9, l. 48: "In the hydrology model, this is represented by parameter $T_c$, which acts to reduce the conductivity as a function of temperature. When the basal temperature is close to the pressure melting point (PMP), there is little change in the hydraulic conductivity. Conductivity decreases to an extremal low value as the temperature approaches the value of $T_c$."

I wonder if the decrease in conductivity could be described by a function? An equation would be very useful here.

**Authors** This was erroraneously stated in the text. The model simply assumes that the frozen bed has a really small conductivty value $10^{-14}$. The text has been modified to include:

> "As a simplifying assumption, the hydraulic conductivity of a frozen grid is set to $10^{-14}$ $\frac{m}{s}$, but can be easily modified to follow a temperature-dependent profile to capture sub-grid variation in basal temperatures."

**Referee** p. 14, l. 12: "The results of these tests show that the model is mass conserving."

This is not very obvious from the rest of the manuscript. I would suggest to add a plot of mass conservation errors (claimed on the order of $10^{12}$ m) or remove this statement.

**Authors** A figure (Figure 2) has been added that shows how well the model conserves mass. The figure shows that it conserves mass on the order of machine precision and any deviations are likely due to rounding errors.

**Referee** p. 1, Affiliations: "St. Johns / St Johns"

The spelling probably needs to be homogenised here.

p. 1, l. 1314: "Channel formation [...] display the arborescent"

An 's is missing here.

p. 7, l. 3031: "As such, there are a number of poorly constrained parameters in the model."

This is a good place to reference to Table 1.

**Authors** These were changed as suggested.

**3   Discussion with Dr. J. V. Johnson**

**Referee** This paper presents a model for subglacial hydrology suitable for continental scale ice sheets. Here, concerns are different from the glacier scale hydrologic models that have seen significant change in recent years. Time steps are longer (days vs hours) and spatial scale is larger (tens of kilometers vs hundreds of meters). The paper is novel in that it defines a set of physical processes related to channelized and distributed flow that can be efficiently and robustly solved using numerical methods. After describing the model, the authors conduct experiments on an idealized, parabolic ice sheet on a flat bed, and conclude with a sensitivity study conducted on a reconstruction of the North American ice sheet complex during the last glacial maximum (18 and 22 kybp).

The model represents worthwhile contribution to the literature because its lower fidelity physics are well suited to the problem ice sheet reconstruction via simulation of 100 ky glacial cycles. Before it is ready for publications, I see a number of issues for the authors to address. I believe that they are significant enough that Ive called them 'major - mostly to assure that something is done to address them. In short, my primary criticism is that I do not think that the results are reproducible because important aspects of model setup are omitted. I also think the work should be better scoped so that readers understand the distinctions between this model and other, recent works related to subglacial hydrology.

**Referee** The simulations, especially those having to do with the LGM (last glacial maximum), have to be described in more detail. Enough is missing that Im struggling to evaluate the conclusions of the paper. In particular: How are model runs set up, and how does the ISM (ice sheet model) get to the point where hydrology is called? I hoped that citing some of the Tarasovs prior work could be used, but didnt find it.

**Authors** First, the code archive (which is now on a public server and which as stated in the submitted text was already available upon request) would enable replication of any of the synthetic ice sheet experiments in the paper.

The North American runs are from full glacial cycle runs starting with ice free conditions at 122 ka. We have inserted the following text: 'The North American ice sheet model is from a large ensemble Bayesian calibration as detailed in Tarasov et al (2012) . Model runs start from 122 ka under ice free conditions."

**Referee** I understand that this is a one-way coupling (ISM forces basal hydro), but that is not sufficient. What are the fields that force the basal hydro model (ice sheet thickness, basal temperature, and basal melt rate?)

**Authors** Dr. Johnson points out that the paper failed to link BrAHMs to the GSM. A paragraph in a new "Model coupling" subsection clarifies the connection between the two models:

> "BrAHMs is highly modular and designed for asynchronous coupling at user specified timesteps. Aside from basic grid information, for each call, the hydrology model requires the following input fields: ice thickness, basal elevation, sea-level, basal ice temperature, basal melt rate, and basal sliding velocity of the ice. For two-way coupling, the relevant outputs from BrAHMS are basal water pressure and thickness."

**Referee** How is the melt-rate computed?

**Authors** From conservation of energy and we don't fathom what other possible option there is. We do not understand why this is being asked given that in the text, we already state: "The GSM is composed of a thermo-mechanically coupled ice sheet model (using the shallow ice approximation), permafrost resolving bed thermal model (Tarasov and Peltier, 2007),"; "The evolving temperature field (T) of the ice sheet is determined from conservation of energy:" eq 1;, "The ice thermodynamics is fully coupled to a 1D (vertical heat diffusion) bed-thermal model"; "Basal temperature is limited to a maximum of the pressure melting point, with excess heat used to melt basal ice". In case this addresses the source of the question, we have replaced the 2nd last quoted sentence above with: "The fully coupled ice and bed thermodynamics are solved via an implicit finite volume discretization in the vertical and explicitly for the horizontal advection component of the ice thermodynamics".

**Referee** Is melt on the surface of the ice sheet routed to the bed?

**Authors** There is no routing though it would be easy to crudely add as a tunable parameter (presumable dependent on ice thickness, longitudinal stress, and ice temperature...). The topic of englacial hydrologic transmission is very non-trivial and beyond the scope of this paper. We have however, added the following text to raise this important issue:

> "For the scope of this initial investigation, we assume no transmission of ice surface melt to the base. Observationally this is known to be false (Zwally et al., 2002), but the dependence on ice thickness, ice temperature profile, and ice strain profile makes this an issue deserving of its own focused study."

**Referee** How is basal traction determined in the absence of two-way coupling?

**Authors** We have inserted the following: "Basal sliding uses Weertman type sliding relations (i.e., function of driving stress) with power law 3 for hard bed and power law 1 for soft bed with sliding onset linearly ramped up starting from $0.2^o\,C$ below the pressure melting point."

**Referee** Finally, see my next point on the stability of a nonlinear system. This is perhaps my greatest concern.

The system of equations includes a number of strong non-linearities in terms of the key prognostic variable - w (effective water depth). Specifically we have Flux, Q, depends on w and K, conductivity, which has w dependence ** Water pressure, or the potential surface that water is routed down, depends on w ** A critical flux, dependent upon w, can have a rapid and strong impact on w.

**Authors** Good point. To address this issue we have added resolution convergence tests using rapidly changing meltwater inputs . The results show convergence and stability (cf new figure 4 and surrounding associated text).

**Referee** What is the interrelation between the time stepping of the ISM and the basal hydrology model?

**Authors** The GSM and BrAHMs are asynchronously coupled. The following is now in the new "Model coupling" subsection:

> BrAHMs is highly modular and designed for asynchronous coupling at user specified timesteps. ...
> Given that there is no lower limit to coupling timesteps, synchronous coupling can also be implemented. For two-way coupling, sensitivity tests are recommended to determine the appropriate coupling timestep for the relevant context.

**Referee** Does the hydrology model achieve steady state between ISM updates? If not, are the larger changes in the potential field on ISM time steps sufficient to produce shocks to the transient hydromodel? Are these shocks captured in a numerically robust way?

**Authors** Cf above note about convergence tests which use basal water supply shocks. Given the asynchronous coupling with user specified timesteps, we see no point in further shock tests (eg change in ice load). Our example results for the North American setup use 5 year timesteps, but 1 (or shorter) year timesteps could also have been carried out and we see no basis for the possibility of a large ice load shock on continental scale ice sheets occuring within a year. The appropriate asynchronous coupling timestep will depend on the modelling context and the revised version of the paper makes it clear that the user needs to carry out sensitivity tests to determine the appropriate timestep.

The numerical stability of BrAHMs comes from both the Heun/Leapfrog-trapezoidal scheme and CFL dependent dynamic timestep setting. The dynamic time stepping that ensures CFL conditions are respected (as already indicated in the abstract). We have also reformulated a relevant section of the text to make this clearer. The beginning of Section 4 now states:

> "The appendix provides detail on the spatial discretization of the equations and the time stepping using the Heun/Leapfrog-trapezoidal scheme. This scheme is second order accurate. The model dynamically adjusts its internal timestep to ensure the CFL criteria is satisfied (with timestep set to $F_{CFL} \times$ minimum CFL timestep). Both of these features contribute to the stability of the model."

**Referee** How does the rapid drainage mechanism, and its impact on the effective pressure, impact the system?

**Authors** The tunnel solver drains the water (the distributed system is paused temporarily... see previous discussion). Once the tunnel solver is finished, the model system continues to run with the new water levels. Since the tunnel solver transports water down hydraulic potentials, hydraulic potential gradients will be reduced under tunnel solver action and therefore numerical stability will not be decreased for the Darcy flow solver (and likely increased).

**Referee** Does it give rise to rapid oscillations in streaming behavior?

**Authors** This is beyond the bounds of this paper. This paper is about the hydrology model, BrAHMS. It does not examine 2-way coupled dynamics. The tunnel solver will evacuate all water from affected grid-cells, so this would shutdown streaming if the basal drag had relevant water pressure dependence.

**Referee** Are any of these non-linear couplings and effects sensitive to the spatial/temporal discretization?

**Authors** The above described convergence tests show minimal sensitivity aside from a low resolution bias that is eliminated in the medium resolution run. Furthermore, the figure 6 sensitivity plots (for North American ice sheet tests) shows no discernable sensitivity to $F_{CFL}$ over a 0.1 to 0.9 range (cognizant that the metric is a bulk measure, ie total water volume).

**Referee** If the ISM is forcing the hydromodel at each ISM time step - then how sensitive is the hydro model to the initial conditions? In particular the distribution of basal water. There are mentions of stability and robust solutions in the text, but they are just that - mentions. Id like to see more on this, to assure the reader the results are stable across discretizations of space and time.

**Authors** The added oscillating water supply convergence test addresses the stability issue. Furthermore, for any glacial cycle modelling, the coupled ISM and basal hydrology would presumably start with the same initial conditions of no water (unless working from a restart file). So we do not understand the point of this question.

**Referee** Continuing with the issue of reproducibility, the code should be more accessible. Publication should include a URL repository where the code can be accessed. Tag the branch used in the publication

**Authors** As noted above, this has now been done.

**Referee** The distinctive features of this model need to be contrasted to the wealth of recent publications in the area of subglacial hydrology. (eg Schoof, Werder, Hewitt, and Hoffmann). There is a need for a continental scale model like this, but it should be established by documenting how and why other models are not suitable to this task. Similarly, the authors claim that other continental scale models do not include sub-glacial hydrology. I dont think this is true. PISM has some accounting for basal water, and so does SICPOLIS. Pollard and DeConto treat hydrology as it relates to sediment. ISSM and Elmer ICE both have hydrology models. CISM contains an ISM, and it contains some subglacial hydrological components developed by Hoffman. It is possible that none of these are good tools for continental/glacial cycle scale studies that are the specialty of Tarasov, but this should be argued persuasively in the paper. Much more should be done here.

**Authors** Clearly, BrAHMs is not the most complex model that exists. It is designed to provide both distributed and channelized flow for large ensemble continental-scale modelling over glacial cycles, and is therefore meant to be fast and robust. The continental scale models listed above (used in CISM and PISM and SICOPOLIS, we could find no description of Pollard and DeConto's treatment), either lack channelized flow or are otherwise too computationally expensive for our stated context.

Dr. Johnson's comments also make clear that our original submission inadequately describe the design context of BrAHMs. Revised and new paragraphs briefly describing these models and clarifying BrAHMs design criteria/context have been added:

> "Subglacial basal hydrology is a potentially critical control on basal drag and therefore ice streaming. It is also a clear control for subglacial sediment production/transport/deposition processes (Benn and Evans, 2010; Melanson, 2012). Subglacial water flows can also leave clear geological imprints. For instance, eskers are a geological footprint of past channelized subglacial drainage (Benn and Evans, 2010) that can in turn be used to better constrain past ice sheet evolution...
>
> Bueler and van Pelt (2015) developed a subglacial hydrology model for the PISM model. Similar to BraHMs, their hydrology model simulates the subglacial water flow using a Darcian flux and limit the basal pressure to the ice overburden pressure (due to long time scales). Their model consists of several basal components, including a water-filled till layer and an effective cavity-based water storage. The model presented by Bueler and van Pelt (2015) does not have any channelized flow mechanisms, which is a major source of water flow/drainage beneath ice sheets, as discussed in Section 2. It is unclear as to how well this model compares to BrAHMs in terms of speed due to the vast difference in model grid and computer usage. The model of Bueler and van Pelt (2015) incorporates the opening and closure of cavities which is necessary for high resolution modelling of present-day glaciers and ice sheets, but can be replaced by low-resolution physics for longer time scales and larger spatial scales, where data is sparse. The incorporation of cavity opening and closure would require computation resources that may be prohibitive for long-term, continental-scale models.
>
> Calov et al. (2018) uses SICOPOLIS in their study on the future sea level contributions of the Greenland ice sheet. The basal hydrology model used in SICOPOLIS is for large-scale grid cells like BrAHMS. The model assumes a thin film of water, resulting in zero effective pressure (meaning the hydraulic potential is simply related to bed elevation and ice sheet thickness). The model first determines a down gradient path from each grid cell to the ocean/boundary. Any depressions are filled with water (akin to lakes) and is given a small gradient so the down gradient solver can continue. From this the water level can be calculated based on the input of meltwater and the hydraulic gradient. This is a rather different approach than BrAHMs as BraHMs attempts to model the physical evolution of the water, allowing for varied flow of water, non-zero effective pressures, and the time-evolution of lake growth under Darcy flow.
>
> Hoffman and Price (2014) also developed a physically-based model to be used as a part of CISM. This model is rather detailed in combining cavity formation (providing water storage) and a method to form Röthlisberger channelized flow. Analysis of

this model looked at fine (100 m x 100m) grids and for shorter periods of time (on the order of days). While they do not mention the speed of their model, given that they model the growth and decay of channels, it is unlikely that the model would be suitable for longer time scales as the time stepping must be small to capture the transient nature of channelized flow. Likewise, for larger grid sizes, the effects of distributed systems (cavities, thin films, etc..) can be averaged out, saving on computation with minimal lost of generality in the results."

**Referee** Axially symmetric experiments should be presented with bivariate plots. Not much is gained by inspecting these highly symmetric solutions. Something is lost in the color map, which might hide high frequency oscillations in the solution.

**Authors** The plots are left unchanged (aside from editing the colours). The aim of these plots were to show the symmetry, as that was the point of the series. The grouping of the colours (as opposed to a continuous scale), was to aid in quickly identifying zones of water height. While not presented in the paper (for brevity), a continuous plot of the data shows no such oscillations exist. The lack of spurious oscillations is even clearer in the now added resolution sensitivity test with time varying meltwater inputs.

**Referee** The focus for the sensitivity study should be streaming behavior, that is the point of the hydro model. Averaging quantities across the entire ice sheet diminishes the importance of changes to parameters. Why not consider the impact of parameter changes to a set of grid cells that are characterized by low effective pressure at 18 and 22 kybp?

**Authors** Our clarification of design criteria hopefully makes it clearer that the point of the basal hydrology model is not just for basal drag (ice streaming), but also for subglacial sediment process modelling and for ice sheet evolution constraint (eg, via tunnel correlation to eskers). Yes, a bulk metric like total water volume hides spatial detail but it still gives a first order measure of model response to parametric uncertainty. Eg, 4 of the 11 parameters have no discernable impact on mean basal water pressure and the net rate of subglacial water evacuation from the ice sheet. We are skeptical that focusing on a few grid cells would be of worthwhile value especially when other uncertainties in interpretation would arise (such as the impact of the cell choice and whether some of the sensitivities might be due to one grid cell changes in flow routing taking flow outside the cell mask).

To our mind, the better sensitivity test would still use bulk measures but a larger set thereof in the context of fully coupled ice sheet and basal hydrology modelling. Feedbacks between the two models could strongly amplify parameter sensitivities hidden by one way coupled studies. This is again beyond the bounds of this study but is the intent of ongoing work in Tarasov's group. The following as been added to the end of the relevant section.

[revised manuscript text omitted]

---

## Author Response (AR2)

**Interactive comment on BrAHMs V1.0: A fast, physically-based subglacial hydrology model for continental-scale application**

Mark Kavanagh and Lev Tarasov

July 21, 2018

**1   Response to single Reviewer (Dr. Johnson) request**

Agreeing with the limited information conveyed in the original map plots of the idealized test cases, we replaced them with transect plots as a function of absolute distance from ice sheet center to portray E-W and N-S symmetry of results. This clearly provides more information value. We thank Dr. Johnson for getting this through our in this case thick skulls.